# Population fluctuations and synanthropy explain transmission risk in rodent-borne zoonoses

Frauke Ecke ®[1,2] ✉, Barbara A. Han ®[3], Birger Hörnfeldt ®[1], Hussein Khalil ®[1], Magnus Magnusson ®[1,4], Navinder J. Singh ®[1] & Richard S. Ostfeld ®[3]

Population fluctuations are widespread across the animal kingdom, especially in the order Rodentia, which includes many globally important reservoir species for zoonotic pathogens. The implications of these fluctuations for zoonotic spillover remain poorly understood. Here, we report a global empirical analysis of data describing the linkages between habitat use, population fluctuations and zoonotic reservoir status in rodents. Our quantitative synthesis is based on data collated from papers and databases. We show that the magnitude of population fluctuations combined with species' synanthropy and degree of human exploitation together distinguish most rodent reservoirs at a global scale, a result that was consistent across all pathogen types and pathogen transmission modes. Our spatial analyses identified hotspots of high transmission risk, including regions where reservoir species dominate the rodent community. Beyond rodents, these generalities inform our understanding of how natural and anthropogenic factors interact to increase the risk of zoonotic spillover in a rapidly changing world.

Rodents are globally abundant and famous for extreme population fluctuations that manifest as boom-and-bust events, eruptive outbreaks and/or cycles[1–5]. The contributors to these fluctuations are heterogeneous, and include density dependence[6], weather conditions[7] that affect food availability[3], predation rates[8], and land use change[9,10], with the importance of these drivers varying across systems[11,12]. Given the near ubiquity of rodents and the diverse ecological roles they play as consumers, prey, and reservoirs for parasites and pathogens, these fluctuations are consequential for many ecological processes, including the transmission of zoonotic pathogens[13–15].

Zoonotic diseases caused by these pathogens are an increasing threat to human health and welfare[16], yet despite our increasing understanding of ecological factors that contribute to outbreaks, our ability to predict zoonotic spillover transmission remains limited. Rodents host a greater diversity of zoonotic pathogens than other

mammal orders[17] and, together with bats and primates, they harbour the majority of zoonotic viruses[18]. Within rodents, species with fast life history strategies (e.g., early and frequent reproduction) appear disproportionately likely to act as zoonotic reservoirs[19], but the mechanisms underlying the effects of life history on reservoir status are poorly understood, as are the pathways leading to direct or indirect contact between rodents and humans[18]—a necessary condition for the transmission of many rodent-borne zoonoses.

The propensity of some rodent species to live exclusively or occasionally in or near human dwellings (synanthropy) has long been acknowledged to increase transmission risk of important zoonoses threatening public health. Synanthropic behaviour and close human contact with globally distributed rodents like the house rat (*Rattus rattus*), brown rat (*Rattus norvegicus*) and house mouse (*Mus musculus*) have been linked to numerous zoonoses, including plague and

[1]Department of Wildlife, Fish, and Environmental Studies, Swedish University of Agricultural Sciences, SE-901 83 Umeå, Sweden. [2]Organismal and Evolutionary Biology Research Programme, Faculty of Biological and Environmental Sciences, University of Helsinki, PO Box 65, FIN-00014 Helsinki, Finland. [3]Cary Institute of Ecosystem Studies, Millbrook, New York 12545, USA. [4]Swedish Forest Agency, Box 284, SE-901 06 Umeå, Sweden. ✉e-mail: Frauke.Ecke@helsinki.fi

typhus[20]. For the remaining majority of rodent reservoirs (282 species), synanthropy is not well documented[21]. In addition to synanthropy, which implies rodents moving into human habitats, contact between humans and rodents can arise from humans moving into rodent habitats (Fig. 1a, b). For example, compared to other professions, forest workers, agricultural workers, hunters and trappers are more frequently exposed to human orthohantavirus infections[22], Lyme disease[22–24], and tularemia[25], with hunters and trappers being at risk either via environmental exposure or due to direct contact with infected animals[26]. As the incidence of zoonoses continues to increase globally, efforts to reduce spillover transmission will depend critically on understanding the mechanisms governing transmissible contact, which are mediated through reservoir traits and ecological context (Fig. 1c).

Here, we augment existing data on rodent reservoirs by compiling key information about rodent population dynamics and degree of synanthropy, which are hypothesized to underlie contact rates with humans across continents. Specifically, we collated or calculated s-index (Methods), which measures the degree of population fluctuation, across

all rodent species for which data were available. This global dataset also includes life history traits, reservoir status for zoonotic pathogens, habitat associations, and spatial distributions of 436 rodent species (Supplementary Data 1). We use these data to test hypotheses about factors characterizing the probability that rodents would pose a strong risk for zoonotic spillover. Specifically, we hypothesised that synanthropic rodents would be overrepresented in the pool of rodent species identified as zoonotic reservoirs. We also hypothesised that zoonotic reservoir species, compared to non-reservoirs, would exhibit large population fluctuations and be habitat generalists. While we predicted that large population fluctuations are an important characteristic of rodent reservoirs, we also hypothesised that exploitation by humans (e.g., by hunting for meat and fur) poses increased transmission risk (Fig. 1). We show that the magnitude of population fluctuations combined with species' synanthropy and degree of human exploitation together distinguish the vast majority of zoonotic rodents at a global scale. Beyond rodents, these generalities inform our understanding of how natural and anthropogenic factors interact to increase the risk of zoonotic spillover in a rapidly changing world.

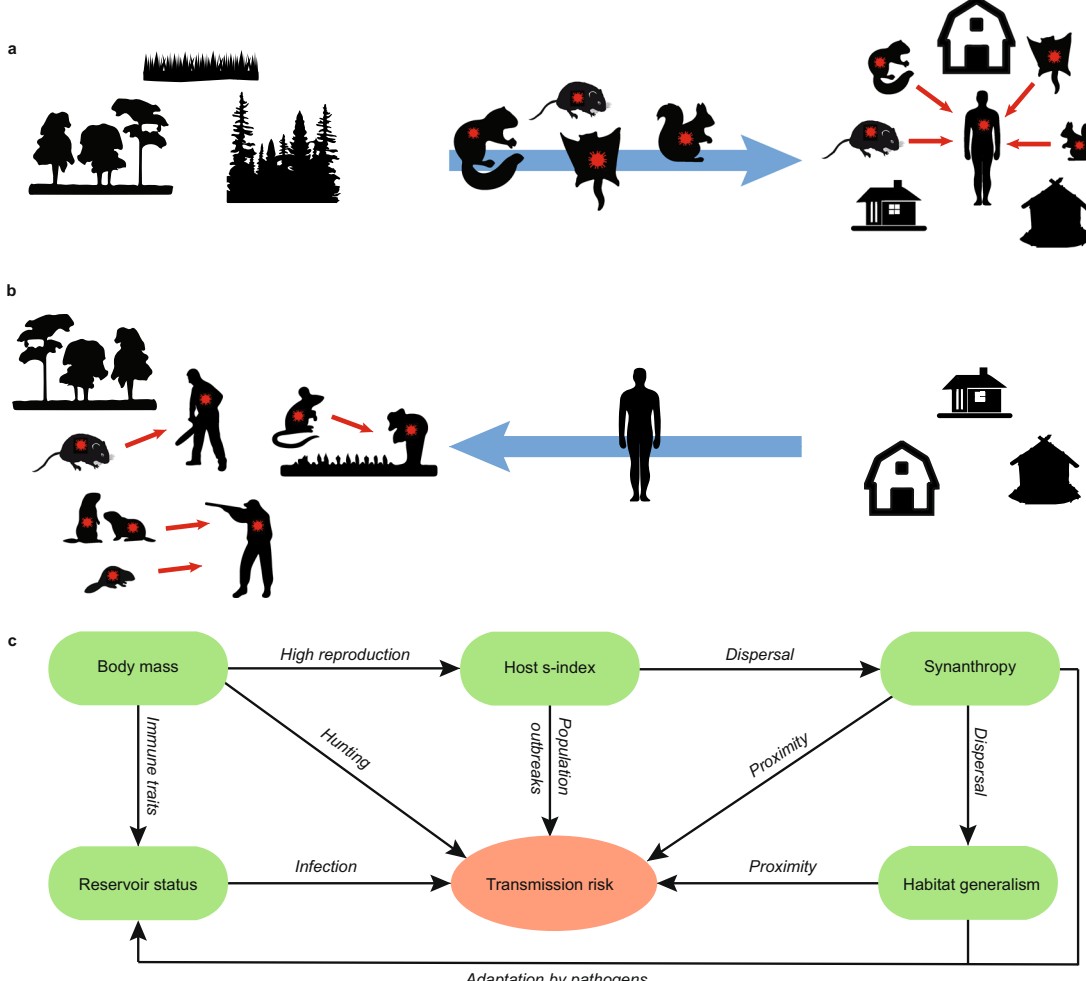

**Fig. 1 | Pathways increasing transmission risk from rodent-borne pathogens.** **a**, **b** Contact between rodents and humans increases transmission risk; **a** either with rodents moving into human dwellings and environments or with **b**, humans moving into rodent habitats or using rodents as a natural resource. **c** Factors (green boxes) and associated key traits of mechanisms (arrows) explaining increased transmission risk from rodent-borne pathogens. Description of mechanisms: Body mass as a key life history trait can dichotomously increase transmission risk. While rodents with high body mass are frequently hunted for fur or meat, many rodents with low body mass have less developed immune defence strategies and exhibit large population fluctuations (high s-index) resulting in population outbreaks. Periods of high rodent population density are frequently associated with abundant dispersal into human-dominated environments. As pathogens are frequently associated with synanthropic and habitat generalist rodents, these rodents increase transmission risk via proximity to humans. Being a reservoir poses an apparent transmission risk. In the light of many rodent-borne pathogens still being undetected[16], factors like synanthropy, habitat generalism, high population fluctuations and/or high body mass are important predictors of transmission risk.

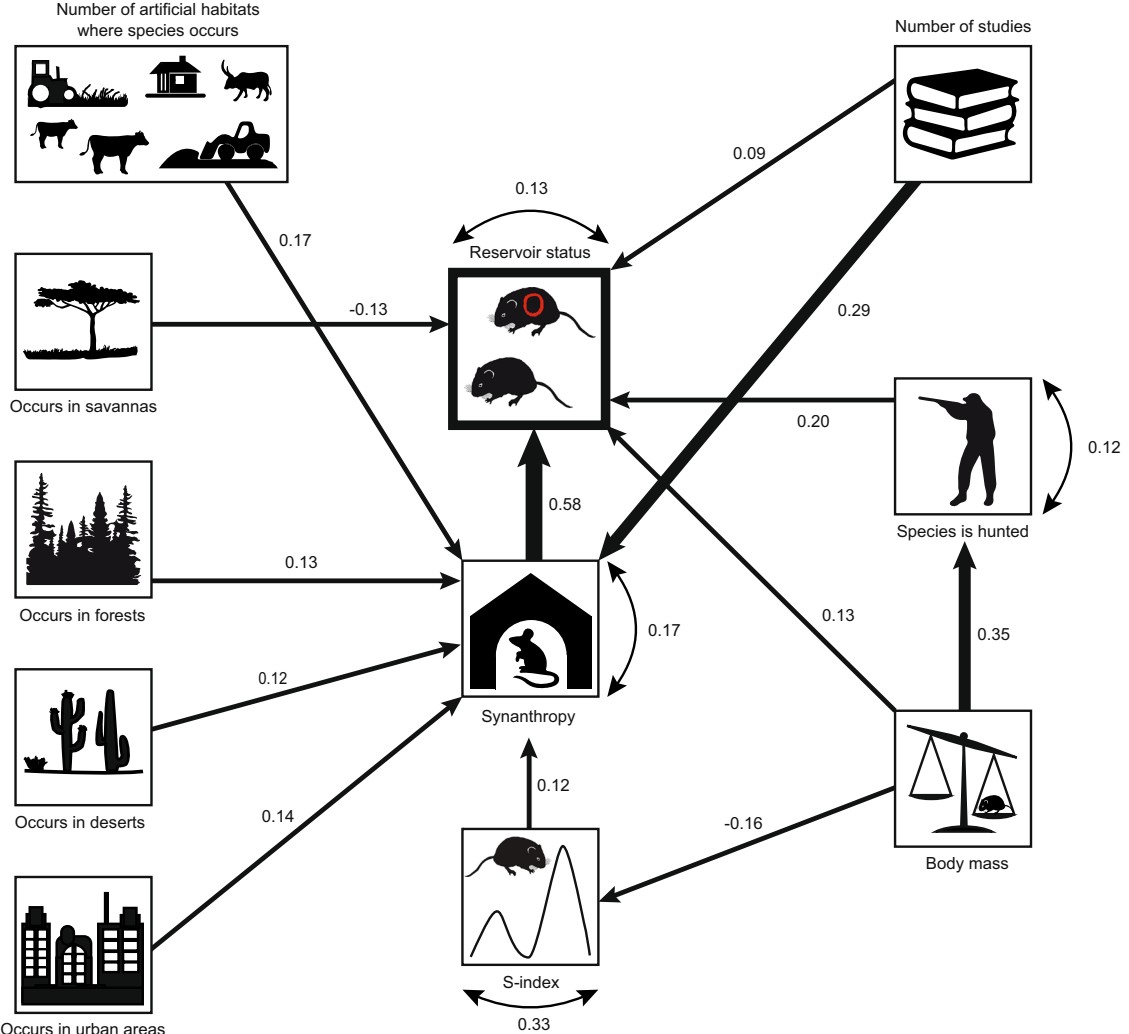

**Fig. 2 | Predictors of reservoir status.** Final structural equation model linking reservoir status of rodent species ($n = 269$) with their synanthropy and hunting status, population fluctuations (s-index, log-transformed), and adult body mass, controlling for their occurrence in a range of habitats and the number of studies available per species. One-sided (directional) arrows represent a causal influence originating from the variable at the base of the arrow, with the width of the arrow and associated value representing the standardised strength of the relationship. The small double-sided arrows and numbers next to each response (endogenous) variable represent the error variance.

## Results

### Predictors of reservoir status

Our analyses include all known rodent reservoirs for zoonotic pathogens (282 species). These reservoirs harbour a total of 95 known zoonotic pathogens (34 viruses, 26 bacteria, 17 helminths, 12 protozoa and six fungi) employing all known modes of transmission (43 vector-borne, 32 close-contact, 28 non-close contact, and 13 using multiple transmission modes) (Supplementary Data 2). Compared to presumed non-reservoirs (species currently not known to harbour any zoonotic pathogens), we observed that reservoir rodents are strikingly synanthropic (Figs. 2, 3a, Table 1). Despite potential geographic biases, and the general possibility that synanthropic species are better studied compared to non-synanthropic species (see Sampling bias and Supplementary Figs. 1, 2), synanthropy emerged as a defining characteristic of nearly all (95%) currently known rodent reservoirs. Of the 155 synanthropic species, only six are considered as truly synanthropic, i.e., predominately, if not exclusively, occurring in or near human dwellings, while the remaining species only occasionally show synanthropic behaviour (Supplementary Data 1).

Compared to non-reservoirs, we also found that rodent reservoirs are disproportionately exploited by humans (hunted for meat and fur).

Seventy-two of the regularly hunted rodent species ($n = 83$) are reservoirs (87%), and hunted rodent species harbour on average five times the number of zoonotic pathogens than non-hunted species (Table 2).

We explored causal pathways using a structural equation model (SEM) linking synanthropy, reservoir status, and their hypothesized predictors. The final model, which we established a priori, had 17 free parameters and 21 degrees of freedom ($n = 269$). The model fit, based on the SRMR (standardized root mean squared residual) and the RMSEA (root mean squared error of approximation) indicated a good fit (see Methods). From the initially formulated full model, the pathways linking reservoir status to population fluctuations (s-index, Methods), occurrence in grasslands, number of artificial habitats a species occurs in, and number of studies found per species were not significant and thus removed from the final model (Supplementary Fig. 3). Similarly, pathways linking synanthropy and occurrence in grasslands were not significant and also removed. All reported coefficients for pathways are standardized to facilitate comparisons among the different relationships. The relationships and coefficients below all refer to those in the final model.

The focal variable in the model was reservoir status, which was strongly and positively associated with synanthropy and had the

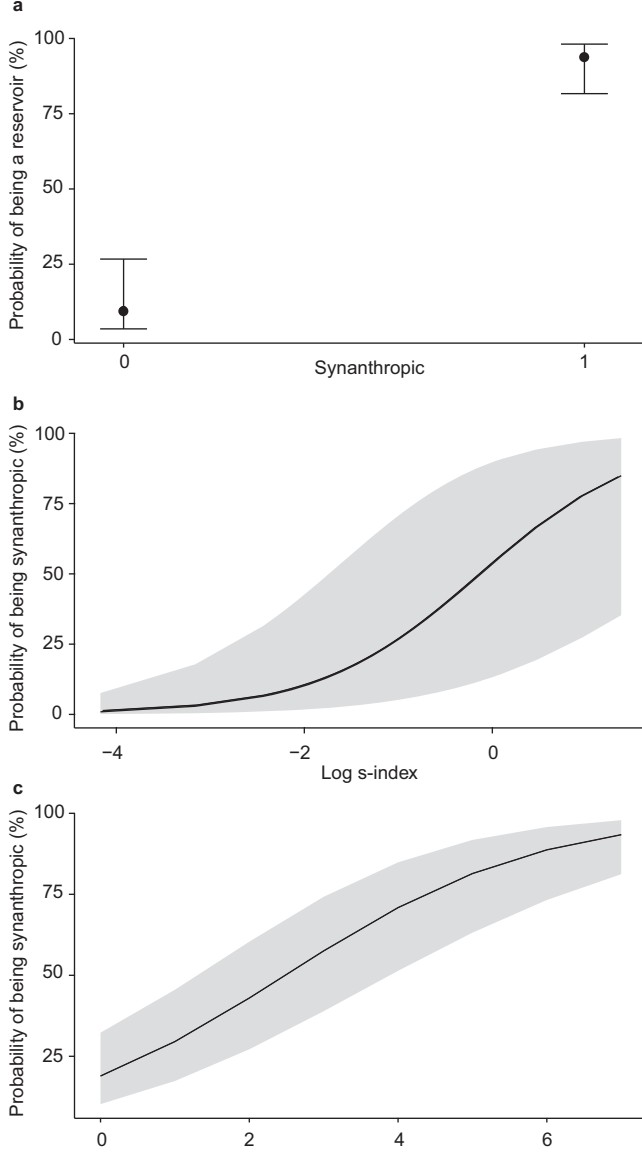

**Fig. 3 | Characteristics of reservoir and synanthropic rodents. a** Reservoir rodents are predominately synanthropic (*n* = 436 with *n* (non-reservoir) = 154, *n* (reservoir) = 282). **b** Synanthropic rodents display high population fluctuations (high s-index) (*n* = 269) and **c**, occur in multiple artificial habitats (*n* = 269) (Tables 1–3). In **a**, estimated probability and 95% confidence intervals are shown and in **b–c**, estimated probability is shown and shaded areas show 95 % confidence intervals.

**Table 1 | Summary of best-fit generalized linear mixed effects model for reservoir status (*n* = 436)**

| Predictors | Odds ratio | 95% CI[a] | *P*-value |
|---|---|---|---|
| Intercept | 0.12 | 0.03–0.51 | 0.004 |
| Log(adult mass, g) | 1.47 | 1.14–1.89 | 0.003 |
| Synanthropic | 132.49 | 58.16–301.83 | <0.001 |
| Occurring in grasslands | 0.37 | 0.22–0.62 | <0.001 |
| Occurring in savannas | 0.43 | 0.21–0.89 | 0.024 |
| Being hunted | 4.65 | 1.83–11.82 | 0.001 |
| Random effects | | | |
| $\sigma^2$ | 3.29 | | |
| $\tau_{00, \text{Rodent family}}$ | 1.15 | | |
| $ICC^b_{\text{Rodent family}}$ | 0.26 | | |
| $N_{\text{Rodent family}}$ | 24 | | |

[a]Confidence interval.
[b]Intra-class correlation coefficient.

The number of studies per species was positively associated with both a species' synanthropic behaviour (0.29, 0.19–0.39) and its reservoir status (0.09, 0.00– 0.19), albeit with weaker evidence for the latter effect (*p* = 0.054) (Fig. 2),

The confirmatory generalized linear mixed effects models (GLMMs) (Tables 1, 3), which control for correlation among species within the same family, showed that our SEM results were robust. Indeed, synanthropy was a significant predictor of reservoir status. These models underscore synanthropy as the most important predictor of reservoir status in our analysis (Table 1, Figs. 2–3).

## Population fluctuations affect transmission risk

Our newly compiled data on the magnitude of population fluctuations enabled comparative investigations beyond theoretically straightforward predictions that transmission risk increases with reservoir abundance for density-dependent systems. We show that while strong population fluctuations (measured as the s-index) are found frequently in both reservoir and non-reservoir rodents (Table 2), synanthropic rodents exhibit much larger population fluctuations compared to non-synanthropic rodents (Table 2, Figs. 2–3). This pattern was apparent despite broad confidence intervals in the relationship between the s-index and the probability of being synanthropic (Fig. 3b, Tables 2, 3). Taken together, our results suggest that larger population fluctuations in reservoir species increase zoonotic transmission risk via synanthropic behaviours of rodents, thereby increasing the likelihood of zoonotic spillover infection to humans.

## Habitat generalism and habitat transformation increase transmission risk

We also find that reservoir species thrive in human-created (artificial) habitats (Fig. 3a, c, Tables 2–3), which reflects a general flexibility in their use of diverse habitat types compared to non-reservoir species (Fig. 4a, Table 2). In addition, the number of zoonotic pathogens harboured by a rodent species increased with habitat breadth ($r_{436}$ = 0.34, *p* < 0.001). Despite the ability to persist in numerous habitat types, we found that reservoir rodents are underrepresented in some natural habitats, especially in savannas and grasslands ($\chi^2$ = 120.81, *df* = 8, *p* < 0.001), and they are overrepresented in artificial habitats ($\chi^2$ = 30.07, *df* = 7, *p* < 0.001; Fig. 4a). Of the 187 rodent species occurring in artificial habitats, 73% are reservoirs, while 59% of the 249 rodents occurring in natural habitats are reservoirs, making artificial habitats more reservoir-rich than natural habitats ($\chi^2$ = 9.28, *df* = 1, *p* < 0.01; Fig. 4a).

Our results also support an emerging consensus that changes in reservoir communities through the degradation of natural habitats

highest estimated pathway coefficient (standardised estimate = 0.58, 95% CI 0.49–0.66, Fig. 2). Controlling for synanthropy, species were more likely to be a reservoir with increasing adult weight (0.13, 0.04–0.22). Species that occur in savanna were less likely to be reservoirs (−0.13, −0.22 to −0.04), while hunted species were more likely to be reservoirs (Fig. 2, 0.20, 0.11–0.30).

Synanthropy was influenced by four habitat variables: a species was more likely to be synanthropic if it occurs in a higher number of artificial habitats (0.17, 0.04–0.31), and occurs in urban areas (0.14, 0.01–0.27), deserts (0.12, 0.01–0.23), or forests (0.13, 0.02–0.24). Notably, species with higher s-index, and thus larger population fluctuations, were more likely to be synanthropic (0.12, 0.01–0.22), and the s-index itself decreased as adult weight increased (−0.16, −0.27 to −0.04). Finally, hunted species were characterized by higher adult bodyweight (0.35, 0.25–0.44) (Fig. 2).

**Table 2 | Summary of rodent characteristics divided by rodent group with respect to hunting, reservoir status, and synanthropic behaviour**

| Rodent characteristics | Mean | 95% CIª | Range | N |
|---|---|---|---|---|
| **Number of zoonoses** | | | | |
| Hunted | 2.40 | 1.88–2.92 | 0–15 | 83 |
| Non-hunted | 0.44 | 0.35–0.53 | 0–3 | 216 |
| **Number of habitats** | | | | |
| Reservoir | 3.26 | 3.00–3.51 | 1–11 | 282 |
| Non-reservoir | 2.87 | 2.58–3.16 | 1–10 | 154 |
| **Number of artificial habitats** | | | | |
| Reservoir | 1.13 | 0.95–1.31 | 0–7 | 282 |
| Non-reservoir | 0.72 | 0.52–0.92 | 0–6 | 154 |
| **Number of habitats** | | | | |
| Synanthropic | 3.96 | 3.56–4.36 | 1–11 | 155 |
| Non-synanthropic | 2.65 | 2.47–2.84 | 1–8 | 281 |
| **Number of artificial habitats** | | | | |
| Synanthropic | 1.60 | 1.31–1.89 | 0–7 | 155 |
| Non-synanthropic | 0.64 | 0.52–0.77 | 0–5 | 281 |
| **s-indexᵇ** | | | | |
| Reservoir | 0.46 | 0.39–0.53 | 0.02–3.94 | 137 |
| Non-reservoir | 0.41 | 0.37–0.45 | 0.11–1.11 | 132 |
| Synanthropic | 0.52 | 0.42–0.62 | 0.09–3.94 | 94 |
| Non-synanthropic | 0.39 | 0.36–0.42 | 0.02–1.11 | 175 |

ªConfidence interval.
ᵇIndex of population fluctuations (Methods).

**Table 3 | Summary of best-fit generalized linear mixed effects model for synanthropic status (n = 269)**

| Predictors | Odds ratio | 95% CIª | P-value |
|---|---|---|---|
| (Intercept) | 0.26 | 0.03–2.10 | 0.205 |
| Number of artificial habitats | 1.96 | 1.58–2.42 | <0.001 |
| Occurring in urban areas | 3.63 | 1.34–9.82 | 0.011 |
| Occurring in deserts | 3.40 | 1.86–6.22 | <0.001 |
| Occurring in forests | 2.84 | 1.71–4.71 | <0.001 |
| Log(s-index)ᵇ | 3.18 | 2.01–5.01 | <0.001 |
| Occurring in grasslands | 0.44 | 0.27–0.72 | 0.001 |
| **Random Effects** | | | |
| $\sigma^2$ | 3.29 | | |
| $\tau_{00,\ Rodent\ family}$ | 2.06 | | |
| $ICC^c_{Rodent\ family}$ | 0.39 | | |
| $N_{Rodent\ family}$ | 19 | | |

ªConfidence interval.
ᵇIndex of population fluctuations (Methods).
ᶜIntra-class correlation coefficient.

increases transmission risk (Fig. 4b). We find evidence that the conversion of natural habitats to human-dominated uses may disproportionately support the persistence of generalist species (Table 2) and facilitate the influx of rodent reservoir species from nearby forest, shrubland and grassland into habitat types in which contact with humans is frequent and zoonotic transmission risk more likely (Fig. 4b).

### Interplay between transmission mode and synanthropy
We examined whether the higher transmission risk imposed by synanthropic species varied with pathogen type or transmission mode. Compared to non-synanthropic species, synanthropic reservoirs harbour a higher number of zoonoses with "close" (transmission via grooming, biting, scratching, aerosols) and vector-borne transmission as the dominant modes (Table 4). The number of zoonoses caused by helminths, bacteria and viruses was also higher among synanthropic reservoirs (Table 4).

### Hotspots of transmission risk
Global analyses of the richness of rodent reservoirs have previously identified hotspots in medium latitude North America and Europe, north-eastern parts of South America, south-eastern coastal Brazil and South-East Asia[18,19]. Our analyses identify additional regions where transmission risk is likely to be high owing to the occurrence of hunted, synanthropic rodents that occupy artificial habitats and show large population fluctuations. We also report regions where reservoir species dominate the rodent community (Fig. 5). These regions include Fennoscandia, South America west of the Andes, southern Australia and New Zealand, where our data suggest that zoonotic risk deriving from rodents is likely to be high because encountering a rodent species largely implies encountering a zoonotic reservoir species (Fig. 5c).

We observed particular regions in which overall rodent richness is low, but the richness of rodent species occupying artificial habitats is comparatively high (Fig. 5d). These regions include the north

temperate zones of both hemispheres (Fig. 5d). In these areas, we postulate that artificial habitats, irrespective of the local species pool, are disproportionately occupied by multiple rodent reservoir species.

### Sampling bias
Generally, the more a rodent species is studied for population fluctuations or zoonoses, the more zoonotic pathogens have been detected in it (Supplementary Fig. 1). The relationship between study effort and pathogen detection, however, is highly variable. For example, two of the five most studied rodent species (*Rattus norvegicus* and *R. rattus*) host 35 and 34 zoonotic pathogens, respectively, while the other three most studied reservoirs (*Mus musculus*, *Myodes glareolus*, and Peromyscus maniculatus) host comparatively few zoonotic pathogens (11, 6, and 10) (Supplementary Fig. 1). In addition, the greater Bandicoot rat (*Bandicota indica*) is a reservoir for the fifth highest number of zoonoses (15), despite comparatively low study effort.

The overall number of studies per rodent species on population dynamics and zoonoses varies among continents ($H_8 = 43.494$, $p < 0.001$, Supplementary Fig. 2) with fewer studies on species occurring exclusively in Africa or Asia compared to those occurring exclusively in North America. However, variation is high and there is a similar number of studies on species occurring in both Africa and Asia compared to species on other continents (Supplementary Fig. 2).

## Discussion
We addressed whether the tendency of some rodents to undergo dramatic fluctuations in population size interacts strongly with habitat generalism, synanthropy, and the tendency to be exploited by humans. These three critical factors combine to predict reservoir status for nearly all known rodent reservoirs of zoonotic disease. It has been long assumed that the nature of the contact interface between humans and reservoir species is important for driving spillover risk. However, our results provide comparative quantitative evidence of the nature of rodent-human contact interfaces for transmission risk, including the encroachment of rodent reservoirs into artificial habitats in addition to human encroachment into natural habitats.

Compared to other mammals, rodents host the highest viral richness of Bunya- and Arenaviruses and, together with bats, the highest number of Flaviviruses[18]. A majority of zoonotic viruses from these families (for revised virus taxonomy see ref. 27) are spread by vectors and/or by close contact (Supplementary Data 2). For such horizontally transmitted pathogens, infection prevalence generally increases concomitantly with reservoir abundance[28]. Periods of rapid

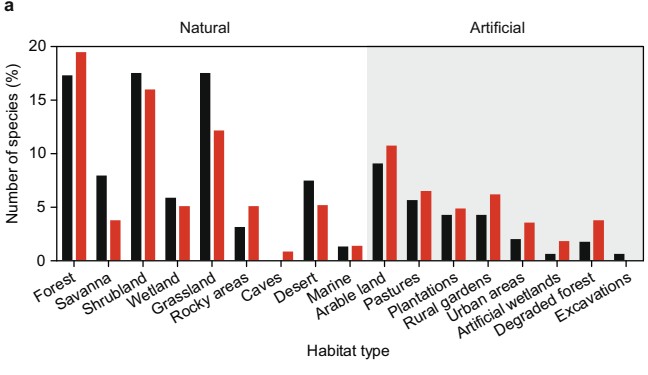

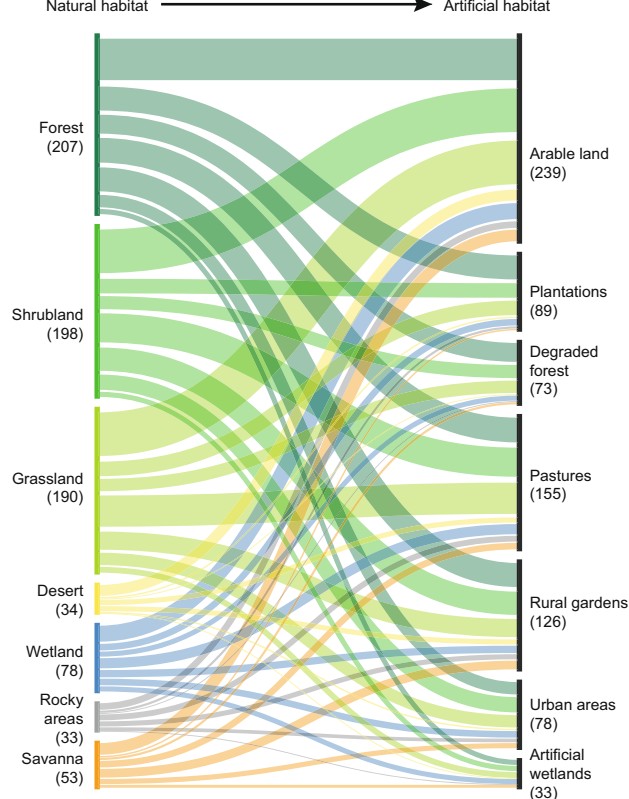

**Fig. 4 | Habitat types occupied by rodents. a** Percentage of non-reservoir (black bars; *n* = 154) and reservoir rodents (red bars; *n* = 282), respectively, occurring in natural and artificial (shaded) habitats. **b** Habitat generalism of reservoir rodents. Reservoir rodents occurring in natural habitats potentially move into artificial habitats, e.g., during periods of high population density or when natural habitat is degraded. Thickness of curves represents number of rodents shared between natural and artificial habitat.

**Table 4 | Summary of number of transmission modes, number of zoonoses with different transmission modes, and number of zoonoses caused by different pathogen types in synanthropic and non-synanthropic rodents**

| Transmission type | Mean | 95% CI[a] | Range | N |
|---|---|---|---|---|
| **Number of transmission modes** | | | | |
| Synanthropic reservoirs | 4.39 | 3.36–5.41 | 1–47 | 148 |
| Non-synanthropic reservoirs | 2.12 | 1.82–2.42 | 1–11 | 134 |
| **Number of zoonoses with close transmission mode** | | | | |
| Synanthropic reservoirs | 1.28 | 1.03–1.54 | 0–10 | 148 |
| Non-synanthropic reservoirs | 0.65 | 0.53–0.77 | 0–3 | 134 |
| **Number of zoonoses with non-close transmission mode** | | | | |
| Synanthropic reservoirs | 0.97 | 0.65–1.28 | 0–15 | 148 |
| Non-synanthropic reservoirs | 0.55 | 0.39–0.71 | 0–5 | 134 |
| **Number of zoonoses transmitted by vectors** | | | | |
| Synanthropic reservoirs | 1.63 | 1.25–2.00 | 0–17 | 148 |
| Non-synanthropic reservoirs | 0.66 | 0.52–0.81 | 0–4 | 134 |
| **Number of zoonoses with intermediate transmission mode** | | | | |
| Synanthropic reservoirs | 0.51 | 0.30–0.71 | 0–9 | 148 |
| Non-synanthropic reservoirs | 0.25 | 0.17–0.34 | 0–2 | 134 |
| **Number of zoonoses caused by helminths** | | | | |
| Synanthropic reservoirs | 0.51 | 0.30–0.73 | 0–10 | 148 |
| Non-synanthropic reservoirs | 0.20 | 0.11–0.29 | 0–3 | 134 |
| **Number of zoonoses caused by bacteria** | | | | |
| Synanthropic reservoirs | 1.18 | 0.88–1.49 | 0–12 | 148 |
| Non-synanthropic reservoirs | 0.36 | 0.25–0.46 | 0–3 | 134 |
| **Number of zoonoses caused by viruses** | | | | |
| Synanthropic reservoirs | 0.99 | 0.78–1.20 | 0–8 | 148 |
| Non-synanthropic reservoirs | 0.38 | 0.29–0.47 | 0–3 | 134 |
| **Number of zoonoses caused by fungi** | | | | |
| Synanthropic reservoirs | 0.03 | −0.01–0.06 | 0–2 | 148 |
| Non-synanthropic reservoirs | 0.05 | 0.01–0.10 | 0–2 | 134 |
| **Number of zoonoses caused by Protozoa** | | | | |
| Synanthropic reservoirs | 0.72 | 0.55–0.90 | 0–7 | 148 |
| Non-synanthropic reservoirs | 0.57 | 0.46–0.69 | 0–3 | 134 |

[a]Confidence interval.

habitats into artificial habitats[33,34] will continue to increase the transmission risk of rodent-borne zoonoses globally (Fig. 4). Our prediction that artificial habitats may be disproportionally occupied by multiple species of rodent reservoirs is supported by multiple previous studies that have shown, at local scales, that habitat loss and fragmentation result in reduced animal diversity[35] with the remaining species dominated by generalists[36] that can reach high density[37]. In addition, complex community responses arise if artificial habitat increases and the population dynamics of native rodents are affected by those of non-native invasive rodents[38]; processes that will likely be more common in the future due to accelerated global change. Our knowledge on the occurrence and distribution of wildlife pathogens, reservoirs and non-reservoirs is constantly increasing, implying that current pathogen-reservoir associations need to be evaluated with caution[39], which also applies to our study (cf. Figs. 4–5). As more rodents will be studied, more pathogens are likely to be detected (Fig. 2)[39].

Complexities of zoonotic transmission, including seasonality of rodent abundance, changes in land use creating artificial habitats, and diversity of transmission modes, cause many zoonotic disease systems to be considered as unique, with a need for control efforts tailored to the ecological nuances of each system. Our analyses demonstrate that across the majority of rodent-borne zoonotic diseases, population fluctuations and associated synanthropy are robust indicators of

population growth to a peak (illustrative of populations with high s-index), can result in rodents dispersing from natural sources to artificial sink habitats including urban areas[4,5,29]. Thus, increases in human transmission risk arise from the interaction between transmission modes, reservoir population fluctuations and habitat generalism in rodent reservoirs.

Our results contradict the perception that natural habitats, due to their high species diversity, pose greater risks of zoonotic transmission compared to artificial habitats[30]. On the contrary, species richness of reservoirs is actually higher in artificial habitats[31], a finding that underscores the high zoonotic risks associated with the conversion of natural into artificial habitats[32]. Since habitat generalists favoured by such land conversions are disproportionally zoonotic reservoir species, our results suggest that the accelerating transformation of natural

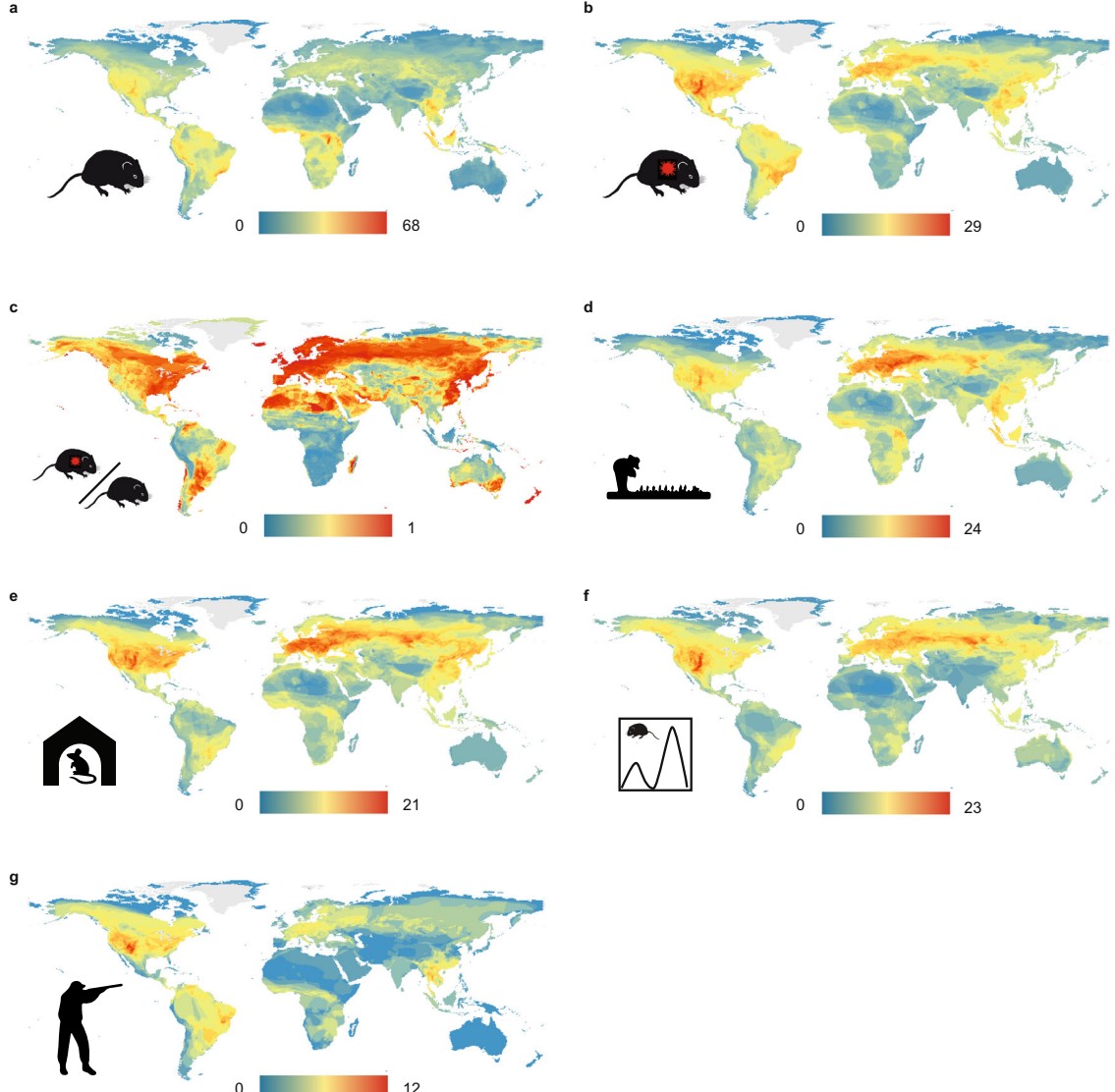

**Fig. 5 | Global distribution of the number of rodent species by category. a** All rodent species (*n* = 2308), **b** reservoir rodents (*n* = 282), **c** ratio between the number of reservoir and total number of rodents, **d** rodents occurring in artificial habitats (*n* = 186), **e** synanthropic rodents (*n* = 155), **f** rodents exhibiting pronounced population fluctuations (s-index >0.3, Methods; *n* = 159), **g** hunted rodents (*n* = 83). Warmer colours highlight areas of high species richness. See Methods for image licensing.

reservoir status and underscore particular pathways to potential spillover transmission. Transmission risk from rodents can however vary within a species' geographic range due to, for example, spatial differences in immunogenetic properties of the reservoir and associated pathogen(s)[40]. The inclusion of reservoir lineages and pathogen strains in future models can therefore provide deeper insight into regional transmission risk.

Given that many rodent-borne pathogens are still undescribed[18] the generalities identified here may serve as useful rules of thumb across rodent-borne zoonoses and also provide useful starting points from which to improve control of zoonotic transmission and more efficient discovery of novel pathogens and reservoir species. To better mitigate disease outbreaks, surveillance focusing on reservoir rodents exhibiting large population fluctuations appears to be a promising approach[13]. Surveillance should in addition intensify screening for new zoonotic pathogens and/or new reservoirs in rodents exhibiting large population fluctuations and those being hunted. These results also suggest the possibility that similar interactions between ecological and anthropogenic factors may exist for other reservoir groups, and may

be leveraged to mitigate increasing risks of zoonotic disease emergence in humans.

## Methods
### Datasets

In our study, we included all rodent species classified as reservoirs of zoonotic diseases by the Global Infectious Disease and Epidemiology Network (GIDEON)[41] by 1 October 2020. Due to the rather recent taxonomic split of *Arvicola amphibius* into *A. amphibius* and *A. scherman*[42], we treated these species as one species complex (here called *Arvicola amphibius*) in the statistical analyses. In total, we studied 282 rodent reservoir species and the nomenclature of all rodents followed the International Union for Conservation of Nature (IUCN) Red List of Threatened Species[43].

We complemented the dataset of reservoir rodents with data for 154 non-reservoir rodent species. These species were selected in a stratified approach. In a first step, we searched in the Clarivate Web of Science © (Copyright Clarivate 2020. All rights reserved) for the most studied rodent species per continent. For this search, we combined

search strings for topics on countries, species (including their synonyms according to IUCN) and information on population dynamics. For information on population fluctuations, we searched for abundance, density, population dynamics, amplitude, cyclic or cyclicity as topics (search finalized 20 May 2020). The search string for each continent is represented as follows: "TS = (country) and TS = (species) and TS = (abundance or density or population dynamics or amplitude or cycl*)", with "TS" representing "Topic", "country" being a list of all countries per continent and "species" being a list of all known rodent species (incl. their synonyms) occurring in the respective continent. We imported the search results into a database and sorted the dataset by the number of references per species. In a second step, we then searched for information on population dynamics and their seasonality of the 20 most studied species per continent (search finalized 19 October 2020). When screening identified articles, we occasionally also found suitable data on population dynamics and/or seasonality for species that were not among the 20 most studied species and included the data in our dataset. For all reservoir rodents, we searched for and extracted literature data on population fluctuations in the same way as described for the non-reservoir species but included also studies listed in Google Scholar. If not available as raw data, we extracted seasonal and/or yearly abundance/density from figures with WebPlotDigitizer[44] or received raw data from the authors. The extracted information per species included minimum and maximum abundance/density, if the respective species exhibits cyclicity, seasonality and/or outbreaks (Supplementary Data 1). For species with data from at least four consecutive years, we also calculated the s-index[45]. This index, calculated as the standard deviation of species abundance or density over time, has frequently been used to quantify the degree of fluctuation in rodent populations[45] and we classified species with s-index >0.3 as those exhibiting pronounced population fluctuations. For species with multiple records for abundance/density, we used the data showing the largest amplitude and the highest value for the s-index, respectively.

As an index of publication bias, we also extracted for each species the number of studies that were available until 1 October 2020 on (a) population fluctuations, (b) zoonoses and (c) population fluctuations or zoonoses from Clarivate Web of Science © (Copyright Clarivate 2020. All rights reserved). For the search on population dynamics, we used the search string "TS = (species) and (TS = (abundance or density or population dynamics or amplitude or cycl*) not TS = (ovulatory cycl*) not TS = (lunar cycl* or lunar phase))", for the search on zoonoses, the search string "TS = (species) and (TS = (zoonoses))" with "zoonoses" representing the list of zoonoses given in Supplementary Data 2, separated by "or", and for the number of studies on population dynamics or zoonoses, we combined the search strings.

Information on habitat preferences of rodent species was extracted from the IUCN Red List of Threatened Species[43]. Forest habitats included woodlands and reforestations, while agricultural fields comprised crop fields, pastures, rural gardens and orchards, and wetlands comprised different types of aquatic habitats that were not used for crop fields. For details on habitat types, see ref. 43. We assigned each habitat type to either natural or artificial habitat (Supplementary Data 1; cf. Fig. 4). Information on synanthropy, i.e., if species live exclusively or occasionally in or near human dwellings, was limited in the IUCN database. We therefore systematically searched for synanthropy in Clarivate Web of Science © (Copyright Clarivate 2020. All rights reserved) (search finalized 19 October 2020) and used the following search terms combined by "or": "synanthrop*", "peridomestic", "village", "domestic", "house", "commensal", "residen*", "human dwelling*", "urban", "infest*", "household*", "anthropogenic" and "outbuilding*" in combination with the respective species names and their synonyms. We also contacted rodent experts to get an expert opinion on synanthropy for rodent species for which we did not find literature data. All species for which neither literature data nor experts

indicated synanthropy, we assigned as non-synanthropic. For all rodent species, we also extracted information on whether they are hunted for fur or meat.

To detect global patterns in the distribution of reservoir rodents, synanthropy, habitat preferences, population fluctuations and hunted rodents, we extracted the distribution range of rodents as vector files from the IUCN[43] database for further use in a geographic information system (GIS). As the IUCN data for introduced species mainly included native distribution ranges, we complemented the distribution ranges of these species (*Cavia porcellus*, *Mus musculus*, *Myocastor coypus*, *Rattus norvegicus*, *R. rattus*, *Sciurus carolinensis*, and *Tamias sibiricus*) with areas they have been introduced to. These data, we extracted from the Global Biodiversity Information Facility (GBIF)[46,47]. To map species richness per hexagon (865 km$^2$), we used the IUCN Species Mapping Tools[48] in ArcGIS Desktop[49].

## Statistical analyses

We used structural equation modelling (SEM) to evaluate the relationship between synanthropy and reservoir status of rodent species, the factors that influence them, and the pathways through which these factors are connected. SEM is a multivariate analysis technique that combines confirmatory factor analyses and multiple regression to test whether the data supports a pre-defined hypothetical model. SEM hypothesizes causal relationships among variables and tests them through linear equations and can include continuous or binomial variables. In SEM, the values assigned by the model to the pathways between variables are standardized estimates of the strength of that relationship after taking into account other relationships specified in the model[50].

Here, we proposed a SEM model (full model) based on our a priori knowledge from previous studies and through preliminary data analysis (cf. Fig. 1). The candidate predictor variables for each model included habitat-related variables, log-transformed s-index (index of population fluctuations), and life history traits (see Supplementary Data 1 for the list of variables). Presented here in Fig. 2, the final model included the s-index, for which data was available for only a subset of the rodent species ($n = 269$). The full model, including all hypothesized pathways, is presented as Supplementary Fig. 3.

After fitting the full model, we removed non-significant pathways ($p > 0.1$). We used RMSEA (root mean squared error of approximation) and SRMSR (standardized root mean squared residual) to test the overall fit of the model. RMSEA is based on overall model $\chi^2$ but is standardized by degrees of freedom and is more appropriate for models with $n > 200$. An RMSEA value of <0.05 indicates a good fit. SRMSR is an absolute measure of fit, defined as the standardized difference between the observed correlation and the predicted correlation, for which a value of <0.08 is considered a good fit[51]. Our final model had a good fit to the data (SRMSR = 0.042). The RMSEA value of 0.071 (90% CI 0.046–0.097) was only slightly higher than the 0.05 suggested cut-off for a good fit. Thus, we proceeded with the interpretation of the final model and the relationships and coefficients therein without further adjustment in the model pathways.

We also fitted two generalized linear mixed effects models (GLMMs) to (a) confirm the results of the SEM and (b) account for the effect of species family on the SEM results and thus correlation among species within the same family, since it is not feasible to include family as a grouping factor in SEM models. The response variables for the two GLMMs were synanthropic and reservoir status, and the candidate predictor variables in the GLMMs were the variables maintained in the final SEM, and we included species family as a random effect.

Each predictor was first tested for its association with the response variable (either synanthropic or reservoirs status) using a generalized linear model with a binomial error distribution, and predictors with a $p$-value < 0.15 were then included in the full model. The full model was a generalized linear mixed effects model with a binomial error distribution. The response variables were reservoir status

(Table 1) and synanthropic status (Table 3). Candidate predictors were included as fixed effects, rodent family as a random effect to control for phylogenetic correlations, and for each species, we weighted the observations by the number of studies with quantitative data included in our dataset (Supplementary Data 1).

Thus, the two models were as follows:

$$Yij \sim \text{Bin}(1, pij) \tag{1}$$

$$\text{logit}(pij|bi) = \alpha + \mathbf{X}\beta + bi + \varepsilon ij \tag{2}$$

$$bi \sim \text{N}(0, \text{D}) \tag{3}$$

$$\varepsilon i \sim \text{N}(0, \sum i) \tag{4}$$

$Y = 1$ if species $j$ in rodent family $i$ was synanthropic (first model) or known reservoir (second model), and 0 if it was not. $\alpha$ is the intercept, $\mathbf{X}$ is a vector of fixed effects, $bi$ is the random intercept for rodent family $i$, and $\varepsilon ij$ is the error component.

We analysed differences in the percentage number of reservoir and non-reservoir rodents, respectively, occurring in natural and artificial habitats by two-tailed $\chi^2$-tests. Correlations between two variables were analysed with Spearman's rank correlation coefficient ($r_s$) and differences among groups with Kruskal-Wallis test ($H$) with Dunn test for posthoc comparisons. Data analyses were conducted in R 4.2.0[52].

### Reporting summary

Further information on research design is available in the Nature Portfolio Reporting Summary linked to this article.

## Data availability

Source data (including rodent species, reservoir status, synanthropy, habitat preferences, population fluctuations, list of reservoir–pathogen associations) needed to fully replicate and evaluate the analyses are provided as Supplementary Information. Global data used for the spatial mapping of rodent occurrences are available from the International Union for the Conservation of Nature and Natural Resources (IUCN, iucn.org) and the Global Biodiversity Information Facility (GBIF, gbif.org).

## Code availability

The model details necessary to replicate the study are given in the Methods section.

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

## Acknowledgements

We thank Janet Foley, Virginia Hayssen, Jens Jacob, Charles Krebs, Herwig Leirs, Yonas Maheretu and Grant Singleton for their expert knowledge on the synanthropy of some rodent species. This study was funded by the Swedish Research Council Formas (Grant No. 2017-00578 and 2017-00867) and the Swedish Environmental Protection Agency (SEPA) through the Swedish Wildlife Management Fund (Grant No. 2020-00093).

## Author contributions

F.E., R.S.O. and B.A.H. designed the study. F.E. and M.M. collected data. H.K. and F.E. performed statistical analyses. F.E., R.S.O., B.A.H., B.H., H.K., M.M., and N.J.S. were involved in writing the manuscript.

## Funding

## Competing interests

The authors declare no competing interests.
