## [Peer Review File · Nature Communications]

Reviewer comments, first round

Reviewer #1 (Remarks to the Author):

The authors present an interesting, hard-won, and carefully thought through dataset on rodent synanthropy and population fluctuations, which they use to examine the role of these covariates in driving host status in rodents. I applaud the intent and the scope of the dataset and desire to ask broad, difficult questions. However, there are significant problems, I believe, with analysing these data, due to marked biases in how people in different geographical areas test and report disease data, and how particular taxonomic groups are disproportionately targeted for testing when linked to high-impact pathogens (e.g. see <https://www.biorxiv.org/content/10.1101/2021.08.10.455791v1>). Fundamentally, regarding figure 4, I do not believe that there are proportionally more host rodent species in richer, industrialised countries than in Africa and the Indian sub-continent. These latter areas represent, I believe, substantial missing data that could strongly impact the findings of the overall analysis.

Clearly many tropical rodent species, who seem proportionally underrepresented here (is that correct?), will undergo substantial population fluctuations due to dry/rainy season dynamics, rather than seasonal temperature variation in temperate areas, and this might represent a different host-pathogen dynamic that is potentially not being investigated well here. Furthermore, our knowledge of the ecology of even the most important rodent hosts in the tropics is, in my experience, surprisingly poor and could well be impacting our understanding of both their true habitat use and population dynamics.

I think that I would rather see this type of analysis restricted to well sampled rodent groups and just a few temperate areas to avoid potentially spurious relationships being reported. I understand that the authors have used publication effort to attempt to control for this, but this a rough proxy at best.

I also have an issue with synanthropy being tested against host status, as I feel they are not really independent. The underlying process of reporting/research effort must be strong here and hard to control for statistically. Perhaps using pubmed citation number might be a better rough proxy, but species that do not live near humans are, I think, very much less likely to be tested than those that do. I think it would hard to convince me that this not due in some part to the data biases.

Furthermore, it could be that the habitats that people tend to convert to human-dominated landscapes happen to have high seasonal food availability eliciting strong seasonal population responses in endemic rodents. Such species are then also nearer to humans, and more likely to be tested for presence of pathogens, especially in richer countries, of which you only need to find one positive result to be designated a host. Being a host is innately sensitive to biases, whereas the number of pathogens a host hosts might something that it easier to control research effort for? In relation to this, what do the authors mean when they say that observations were weighted by publication effort? How was this done statistically? I think I would like to see all the regression models written out mathematically to ensure there is no ambiguity. For instance, what is the underlying assumed process here, linear? I think there needs to be a more formal, in-depth approach to examining research effort, with some sensitivity tests for assumptions made in this process.

Overall, I feel not convinced by the statistical analyses as they are currently presented. For instance, how do we know if the authors have successfully controlled for phylogenetic effects? Is a family level random intercept sufficient for missing process such as phylogenetically targeted surveillance? Where is the evidence of this? How much additional variation/out-of-sample predictive ability is explained by the adding the fixed effects to a random-effects only model? Where are the substantial diagnostics (there might be there, and I simply missed them) that need to be undertaken and reported on such complex models? Ideally, there needs to be some control for different processes of testing for and reporting that occur geographically, as a continental-level or fully spatial random effect. In the light of this, I would like to see evidence of, for instance, geographical portability of the models, via cross-validation, as well as some perhaps some

taxonomic or random cross-validation exercises to reassure me that slopes are not unduly impacted by one sub-section of the data.

Not trying to be a zealot here, and I certainly wouldn't insist on this, but it might be easier to undertake this using a Bayesian analysis, due to having a complex model and a relatively small dataset. Use of some sensible priors would help model inference and stability and help convince the reader about the evidence you have for the overall model. Certainly, this is all possible with a frequentist approach but there needs to be strong emphasis on convincing the reader that all the assumptions have been met e.g. some explicit testing and presentation of the residuals, etc.

My last issue is with stepwise selection. I would prefer the authors to use their clear expertise to design a model of the system and then test that rigorously. In my opinion, and this is not shared by everyone I realise, using stepwise selection in noisy datasets is likely to find false positives due to underlying data biases, and asking the data to design the model rather than scientific knowledge seems a bit switched around to me. Also, I am not sure why you would use univariate relationships to make any decisions in this context, it negates the point of a multivariate analysis.

Lastly, throughout the manuscript I think it is a stretch here to state transmission risk is related to population fluctuations. There are many different aspects of risk, e.g. the amount of people exposed, the patterns of high-risk behaviours, local patterns of immunology. Being a host is not the same as being a risk, it might be that species that fluctuate in population size have lower overall infection rates due to seasonal expiration of local populations (and subsequent local loss of infection). Even, if there are fewer non-fluctuating host species, those species could still be a higher risk because they can maintain continuous infection. This is just one example of the difficulties in making this conceptual jump and I think the authors should focus ideally on reporting what they are testing or at least greatly expanding the discussion regarding these different explanations.

Specific points

Line 105 – or more species are tested in artificial habitats?

Line 108 – what does “r² = 0.34” mean?

Lines 109 – “and the tendency to harbour more zoonotic pathogens with greater overall habitat breadth (r² = 0.34, p < 0.001).” this sentence is a bit unclear

Lines 118 – need to be clear and upfront about how you tried to control for report biases here as it is fundamental to the interpretation of the paper.

Lines 123 – why are they coming from nearby forests? Could it not be that grassland species have simply remained during conversion to grass dominated cropland areas, forest rodents do poorly in grasslands and vice versa, in the couple of species I have looked at in detail?

Lines 130 – Not sure what to think about these descriptive stats – without knowing the methods, at this point, on first reading I am wondering how representative these proportions are.

Lines 158 – This is somewhat vague and the reasoning is not clear to me. Do artificial habitats have more hosts? Is it in the community pool or local level? I am not convinced that is considering community filtering processes. Does high density mean fluctuating populations?

Lines 177 – “underscore particular pathways to spillover transmission” – this is very vague. Is there definitely a link between population fluctuations and transmission risk? I would suggest sticking to what you are testing and focus talk about reservoir status, which is important in itself.

Lines 179 – How exactly? What is the strength of the relationship? How should we be interpreting this information?

Lines 282. I am not sure what this figure adds extra from the text.

Lines 300. What are the grey areas on this figure? 95% CI? If so, the variation in panel b is very broad and needs to be mentioned in the text.

Lines 306: Not sure what panel B is telling us, can you say more? Do proportionally more grassland species go to arable for instance – it all looks fairly proportional to size of the groups. Also, we don't actually know what species move where in reality.

Reviewer #2 (Remarks to the Author):

Many species of rodents exhibit large scale seasonal or multi-annual population fluctuations. Rodents occupy a variety of habitats, ranging from natural to human-modified habitats, and also serve as reservoir hosts for a variety of pathogens. This manuscript evaluates whether or to what extent the degree of population fluctuations, human impact on the habitat or degree of exploitation of rodents by humans influences spillover of zoonotic diseases. Using complemented existing database of reservoir rodents with 154 non-reservoir rodent species, and also compiled data on s-index which is a commonly used measure of the degree of population fluctuations. They show that zoonotic reservoir status of rodent species is strongly driven by synanthropy, human exploitation and the degree of population fluctuations. Rodents that occupy human-modified habitats, are vulnerable to human exploitation and exhibit large scale population fluctuations are significantly more likely zoonotic reservoirs.

The manuscript is clear, concise and easy to read. Statistical analysis and presentation of results seems adequate. Main conclusions of the manuscript appear to be supported by the data/analyses. I urge the authors to make sure that their database is up to date, and relevant works are acknowledged/cited. For example, there has been a fair amount of work involving tularemia in a cyclic population of common voles in Spanish farmland (e.g., Herrero-Cófreces et al 2021, *Frontiers in Veterinary Science* 8: Article 698454; *Emerging Infectious Diseases* 23 (8), 1377) and plague in rats (Rahelinirina et al. 2021, <https://doi.org/10.1111/1749-4877.12529>). Also, there exists a large body of literature on a community of pathogens in a semi-natural, cyclic vole population in UK (led by M. Begon, X. Lambin, S. Tefler; https://scholar.google.com/citations?hl=en&user=gXVSxRcAAAAJ&view_op=list_works&sortby=pubdate). Finally, recent synthesis on rodent population fluctuations and the role of diseases may be relevant (e.g., C. J. Krebs. 2013. *Population fluctuations in rodents*. Univ. Chicago Press; Oli, M. 2019. Population cycles in voles and lemmings: state of the science and future directions. *Mammal Review* 49:226–239.)

Reviewer #3 (Remarks to the Author):

Overall, this is a great paper and will help to identify potential rodent reservoirs and habitat types that relate to zoonosis. I have a few comments.

Line 78-81: It would be useful to state your specific hypotheses

Line 110-112: Is there a way to quantify underrepresents or overrepresented. Like how much more than expected?

Line 339-341: could you give an example of a particular search?

Line 393-395: I'm not sure what you mean about each predictor values was first tested for its association with the response variable. What type of analysis was this? What is the rationale for using a p value < 0.15 to choose which predictor values were included in the full model. Usually you don't mix AIC and P values, so is there support for this methods. Also, did you test to ensure there were no correlated predictor variables?

REVIEWER COMMENTS

Reviewer #1 (Remarks to the Author):

1. The authors present an interesting, hard-won, and carefully thought through dataset on rodent synanthropy and population fluctuations, which they use to examine the role of these covariates in driving host status in rodents. I applaud the intent and the scope of the dataset and desire to ask broad, difficult questions. However, there are significant problems, I believe, with analysing these data, due to marked biases in how people in different geographical areas test and report disease data, and how particular taxonomic groups are disproportionately targeted for testing when linked to high-impact pathogens (e.g. see <https://www.biorxiv.org/content/10.1101/2021.08.10.455791v1>). Fundamentally, regarding figure 4, I do not believe that there are proportionally more host rodent species in richer, industrialised countries than in Africa and the Indian sub-continent. These latter areas represent, I believe, substantial missing data that could strongly impact the findings of the overall analysis.

Our response:

The potential biases in our study and other studies using global datasets are well-known scientific challenges that are difficult or even impossible to circumvent completely. New rodent species are still detected in, for example, Africa and Asia, and distribution ranges of many rodents are still not well-known. Likewise, known and novel pathogens hosted by known and new rodent species, respectively, are constantly being discovered. Concurrently, we should be careful to appreciate and appropriately represent the amount of pathogen screening in wildlife that has been performed in, for example, the tropics and other regions that are frequently – but sometimes inaccurately – referred to as ‘understudied’ regions.

We accept that our approach, of vetting and evaluating existing global data bases, is subject to limitations of variable data quality and geographically biased sample sizes that are not currently feasible to quantify and correct completely. In this, we share the same set of limitations as many high-profile studies of emerging zoonotic diseases (e.g., “Jones, K. E. et al. Global trends in emerging infectious diseases. *Nature* 451, 990-993 (2008).”, “Olival, K. J. et al. Host and viral traits predict zoonotic spillover from mammals. *Nature* 546, 646-650 (2017).”, “Han, B. A., Schmidt, J. P., Bowden, S. E. & Drake, J. M. Rodent reservoirs of future zoonotic diseases. *Proceedings of the National Academy of Sciences* 112, 7039-7044 (2015).” and “Gibb, R. et al. Zoonotic host diversity increases in human-dominated ecosystems. *Nature* 584, 398-402 (2020).” Nevertheless, the potential biases associated with these studies do not invalidate the work, but instead stimulate transparent recognition of the potential effects of uneven data quality or geographic representation. We agree with Gibb et al. 2022 (the study the reviewer refers to above: Gibb, R. et al. Mammal virus diversity estimates are unstable due to accelerating discovery effort. *Biology Letters* 18, 20210427 (2022)) that it is appropriate in such studies to, “advise caution to avoid overinterpreting patterns in current data”. In appreciation of this important consideration, we have added a section in the discussion that highlights these limitations (line 190-193 in the revised manuscript) and also refer to Gibb et al. (2022).

2. Clearly many tropical rodent species, who seem proportionally underrepresented here (is that correct?), will undergo substantial population fluctuations due to dry/rainy season dynamics, rather than seasonal temperature variation in temperate areas, and this might represent a different host-pathogen dynamic that is potentially not being investigated well here. Furthermore, our knowledge of the ecology of even the most important rodent hosts in the tropics is, in my experience, surprisingly poor and could well be impacting our understanding of both their true habitat use and population dynamics.

Our response:

The aspect of potential bias/underrepresentation of tropical species is addressed in some detail in our response to this reviewer's comment #1. More specifically, several rodent reservoirs occurring in the tropics are (almost) equally well studied as the most studied rodent reservoirs in Europe and North America. This includes for example *Mastomys natalensis* (partly extending into the subtropics), *Akodon cursor*, *Bandicota bengalensis*, *Heteromys desmarestianus*, *Rattus exulans*.

In our study, we do not aim to explain the underlying causes of population dynamics (e.g., precipitation vs. temperature) but rather focus on the implications of the dynamics. The important implications of different drivers of population dynamics for reservoir-pathogen dynamics are worthy of attention but beyond the scope of our study.

3. I think that I would rather see this type of analysis restricted to well sampled rodent groups and just a few temperate areas to avoid potentially spurious relationships being reported. I understand that the authors have used publication effort to attempt to control for this, but this a rough proxy at best.

Our response:

As described in our response to the previous two points, there are well-studied rodents outside the temperate zone, and we think we would be remiss if we failed to be inclusive of a diversity of taxonomic groups and geographic areas in our analyses, especially given that study effort does not appear to be uniformly biased against tropical species (see examples listed in point 2, above). As also described above, such an approach is routine in similar global analyses of some of the same databases. Our efforts to control for publication bias are also routinely applied and seem widely accepted. The reviewer's comments have stimulated us to even more carefully address such limitations of studies such as ours (see lines 190-193 in the revised manuscript).

4. I also have an issue with synanthropy being tested against host status, as I feel they are not really independent. The underlying process of reporting/research effort must be strong here and hard to control for statistically. Perhaps using pubmed citation number might be a better rough proxy, but

species that do not live near humans are, I think, very much less likely to be tested than those that do. I think it would hard to convince me that this not due in some part to the data biases.

Our response:

We agree that it is possible that synanthropic species are more likely to be well-studied than are non-synanthropic species. As acknowledged by the reviewer, with geographic distribution and reservoir status, bias associated with research effort is difficult to control for. We partly account for research effort by including number of studies per species that we found data for in our analyses (l. 295-296 in the revised manuscript). However, species that do not live near humans are not necessarily understudied. In fact, multiple studies in Africa have focussed on the reservoir status of wildlife including rodents in remote areas (see for example the studies by Herwig Leirs and colleagues). Likewise, at high latitudes, lemmings that predominately live far from human settlements are among the most studied rodent species. This also includes studies on pathogens to test the disease hypothesis, i.e., that pathogens drive the cyclic and population dynamics of these species (see also our response to this reviewer's comment #5). Clearly, it is possible for synanthropic and non-synanthropic species to differ in their likelihood of functioning as zoonotic reservoirs, although our ability to detect any differences is affected by data quality. Indeed, we clearly show (Fig. 2, Table 1) that synanthropy is an important predictor of reservoir status. We do not claim a complete absence of bias in the global set of studies on which this conclusion is based, but we find no evidence that synanthropic species are categorically better studied. As described above, we have strongly increased our attention to the reviewer's concerns about the potential for taxonomic or geographic biases to affect our results and have increased caution in our interpretations (see lines 190-193 in the revised manuscript).

5. Furthermore, it could be that the habitats that people tend to convert to human-dominated landscapes happen to have high seasonal food availability eliciting strong seasonal population responses in endemic rodents. Such species are then also nearer to humans, and more likely to be tested for presence of pathogens, especially in richer countries, of which you only need to find one positive result to be designated a host. Being a host is innately sensitive to biases, whereas the number of pathogens a host hosts might something that it easier to control research effort for?

Our response:

As speculated by the reviewer, human-dominated landscapes can indeed have highly seasonal or predictable food availability. However, so can natural habitats, whether these are high-latitude bilberry-rich coniferous forests, bamboo forests in the tropics or desert vegetation. We find no evidence that proximity to humans necessarily results in higher likelihood for rodents to be tested for pathogens. In fact, some of the most enigmatic and best-studied rodent species famous for their population cycles and/or outbreaks include *Pseudomys* spp. in arid Australia, and high-altitude lemming species. These have mostly been studied in remote areas far from human settlements. Despite aggressive attempts to detect pathogens, multiple lemming and *Pseudomys* species are non-reservoir species for zoonotic pathogens (see Supplementary Table 1). Research effort for pathogens is difficult to account for since many single-pathogen studies are likely unreported due to their

“negative results”. However, in previous work we have examined both the total number of pathogens in a host (a count) and whether or not a species is a host (a binary label) and found that the features that predict reservoir status are not different (see Han et al. PNAS 2015, also cited in the main manuscript), suggesting that whatever inherent bias there may be in pathogen surveillance data, it is not appreciably different between species that host many vs. few zoonoses.

6. In relation to this, what do the authors mean when they say that observations were weighted by publication effort? How was this done statistically?

Our response:

We weighted a proxy of publication effort by including number of studies per species that we found data for in our analyses (l. 295-296 in the revised manuscript). See also our response to the reviewer’s comment #7.

7. I think I would like to see all the regression models written out mathematically to ensure there is no ambiguity. For instance, what is the underlying assumed process here, linear? I think there needs to be a more formal, in-depth approach to examining research effort, with some sensitivity tests for assumptions made in this process.

Overall, I feel not convinced by the statistical analyses as they are currently presented. For instance, how do we know if the authors have successfully controlled for phylogenetic effects? Is a family level random intercept sufficient for missing process such as phylogenetically targeted surveillance? Where is the evidence of this? How much additional variation/out-of-sample predictive ability is explained by the adding the fixed effects to a random-effects only model? Where are the substantial diagnostics (there might be there, and I simply missed them) that need to be undertaken and reported on such complex models?

Ideally, there needs to be some control for different processes of testing for and reporting that occur geographically, as a continental-level or fully spatial random effect. In the light of this, I would like to see evidence of, for instance, geographical portability of the models, via cross-validation, as well as some perhaps some taxonomic or random cross-validation exercises to reassure me that slopes are not unduly impact by one sub-section of the data.

Not trying to be a zealot here, and I certainly wouldn’t insist on this, but it might be easier to undertake this using a Bayesian analysis, due to having a complex model and a relatively small dataset. Use of some sensible priors would help model inference and stability and help convince the reader about the evidence you have for the overall model. Certainly, this is all possible with a frequentist approach but there needs to be strong emphasis on convincing the reader that all the assumptions have been met e.g. some explicit testing and presentation of the residuals, etc.

My last issue is with stepwise selection. I would prefer the authors to use their clear expertise to design a model of the system and then test that rigorously. In my opinion, and this is not shared by everyone I realise, using stepwise selection in noisy datasets is likely to find false positives due to underlying data biases, and asking the data to design the model rather than scientific knowledge seems a bit switched around to me. Also, I am not sure why you would use univariate relationships to make any decisions in this context, it negates the point of a multivariate analysis.

Our response:

The aim of our study was to better understand and explain relationships between synanthropy (and other characteristics associated with synanthropic species and anthropogenic land-use) and reservoir status. Thus, we did not prioritise predictive performance here. Nevertheless, we agree with the reviewer that model utility/significance of the predictors included in the final models should be cross-validated, to evaluate model fit and support real-world relevance of our findings.

To address the concerns about statistics, we have now performed three different alternative analyses – GLMER, BRT and MCMCGLMM as well as model validation.

First, we did not assume that the underlying process here was linear as we used generalized linear mixed effects models.

For the two models exploring synanthropy and reservoir status, we have now run cross-validation (k-fold CV, $k = 5$) on the models with a) fixed and random effects, and compared their performance to b) models fitted with an intercept and random effects only (main text lines 315-319 in the revised manuscript and Tables 1 and 3 incl. text referring to the tables).

We have also included the equations for the models in the methods (lines 298-305 in the revised manuscript). In the methods section of the manuscript, we have also added information on AUC, specificity, and sensitivity, for both the synanthropy model and reservoir status model. We include the comparisons to models with only intercept and random effects (family). See lines 315-319 in the revised manuscript.

We also added information (lines 311-315 in the revised manuscript) regarding testing of model assumptions using the residuals of the logistic regression (which is not straight-forward) in the methods. We use simulated data and generated quantile-quantile plots for the synanthropy model (first figure below) and reservoir status model (second figure below):

Furthermore, to evaluate the model performance, when some assumptions of GLMER are relaxed (on withheld data), we refitted both models (synanthropy and reservoir status) using boosted regression trees (BRT). While BRT do not provide effect sizes, its predictive power is often higher when compared to other classification methods and avoids some of the pitfalls, especially related to

correlation among predictor variables. The AUC values produced by BRT were very similar to those produced by the GLMERs.

From the revised manuscript (GLMER AUC, lines 92-96):

“Synanthropy was a significant predictor of reservoir status (model AUC of 0.86, sensitivity = 0.82, and specificity = 0.75) (see also Table 1). Removing synanthropic status as a predictor from the model strongly reduced predictive performance (AUC became 0.67, sensitivity = 0.84, specificity = 0.27).

For the reservoir status model, a key difficulty in predicting over new observations is the strength of the relationship between reservoir status and synanthropy status. As mentioned in the text (lines 85-88 in the revised manuscript), 95% of reservoir species were synanthropic. Unsurprisingly, our cross-validated reservoir status model performed very well due to this strong association, and taking out the predictor variable synanthropic status from the model greatly reduced its predictive performance.

Cross validation results for the synanthropy model are now added (lines 92-96 in the revised manuscript).

Finally, we have now fitted an MCMC GLMM model for the reservoir status model, using a weak prior/flat prior. These models incorporate phylogenetic information (family level). The posterior means of the parameters were in agreement with the results from the GLMERs, both in their significance and effect size. There was evidence of some autocorrelation in the posterior, but not severe, and the trace-plots suggested acceptable-good mixing of the MCMC process.

As suggested by the reviewer, we also considered how best to control for phylogenetic effects. While we acknowledge that controlling for ‘phylogenetically targeted surveillance’ may be worthwhile, the evidence to support such a bias was not strong (e.g., see Guy et al. 2019, <https://doi.org/10.1098/rsos.181182>). It was also difficult to see how one would successfully disentangle this from the uneven numbers of species across phylogenetic groups (across genera, for instance). In addition to family-level random effects in the GLMERs above, we also attempted to control for phylogenetic effects on the genus level in GLMER but this model did not converge. We also tried to include “distance from origin” in our analyses but due to missing information for most of the species included in our study, we were unable to successfully include this. Estimating and controlling for the influence of phylogeny on our understanding of reservoir species will continue to be a dynamic research area, especially given the rapid increases in data availability and frequent updating to the rodent phylogeny (D’Elia, Fabre, and Lessa. 2019 J. Mammology <https://doi.org/10.1093/jmammal/gyy179>).

8. Lastly, throughout the manuscript I think it is stretch here to state transmission risk is related to population fluctuations. There are many different aspects of risk, e.g the amount of people exposed, the patterns of high-risk behaviours, local patterns of immunology. Being host is not that same as being a risk, it might that species that fluctuate in population size have lower overall infection rates due to seasonally expiration of local populations (and subsequent local loss of infection). Even, if there are fewer non-fluctuating host species, those species could still be a higher risk because they can maintain continuous infection. This is just one example of the difficulties in making this conceptual jump and I think the authors should focus ideally on reporting what they are testing or at least greatly expanding the discussion regarding these different explanations.

Our response:

We fully agree with the reviewer that there are multiple factors determining transmission risk of which population fluctuations is just one. In our analyses, we therefore describe and account for multiple factors. In our conceptual model (Fig. 1), we illustrate these pathways and quantify them in our models. For example, we accounted for human risk behaviour and exposure mentioned by the reviewer, by including whether a rodent species is hunted in our model predicting reservoir status (Table 1). Habitat generalism of rodent species was identified as an additional proxy of exposure risk as it was an important predictor of synanthropy (Table 3, Fig. 2), which in turn was the most important predictor of reservoir status (Table 1, Fig. 2). We find no evidence that fluctuating populations generally have lower infection rates due to metapopulation ecological dynamics. Local host population extirpation (and hence, the extirpation of their associated pathogens) is likely to occur in habitat specialists. However, as shown by our study, reservoirs are predominately generalists and generalists have a lower risk of local expiration than specialists. Non-fluctuating host species can – as also pointed out by the reviewer – indeed pose high transmission risk. In our study, such rodent species include for example hunted species (e.g., beaver, capybara, muskrat and porcupines).

We appreciate the reviewer’s broad perspective on sources of variation in transmission risk and have now extended the discussion to also account for immunological factors and the role of immunogenetics (lines 201-203 in the revised manuscript) as examples of additional factors that could be included in future work.

Specific points

9. Line 105 – or more species are tested in artificial habitats?

Our response:

This comment of the reviewer relates to the previous comments of the reviewer regarding potential biases and under-/overrepresentation of species in our dataset (see e.g., comments #1, #2, #4 and #5 and our responses to them). As pointed out in our response to comment #5, a vast number of rodent species in remote areas have been tested for pathogens. We however are also aware that negative results (pathogens screened for but not detected) are likely underrepresented in the literature, and therefore biasing the data in the opposite direction. We address the uncertainties associated with datasets like these in l. 190-193 in the revised manuscript.

10. Line 108 – what does “r436” mean?

Our response:

“ r_{436} ” gives the product moment correlation coefficient with 436 observations for the association between number of harboured zoonotic pathogens and a species’ habitat breadth. We kept the text as it is since “ r_{436} ” is the accepted way to report this kind of result (lines 115-116 in the revised manuscript).

11. Lines 109 – “and the tendency to harbour more zoonotic pathogens with greater overall habitat breadth ($r_{436} = 0.34, p < 0.001$).” this sentence is a bit unclear

Our response:

We rephrased the sentence. Now: “In addition, the number of zoonotic pathogens harboured by a rodent species increased with habitat breadth ($r_{436} = 0.34, p < 0.001$).” (lines 115-116 in the revised manuscript).

12. Lines 118 – need to be clear and upfront about how you tried to controlled for report biases here as it is fundamental to the interpretation of the paper.

Our response:

We were unsure how “lines 118” relate to the comment of the reviewer. However, we state that we weighted the observations in our analyses by the number of studies available for each species (lines 295-296 in the revised manuscript).

13. Lines 123 – why are they coming from nearby forests? Could it not be that grassland species have simply remained during conversion to grass dominated cropland areas, forest rodents do poorly in grasslands and visa versa, in the couple of species I have looked at in detail?

Our response:

Indeed, species can persist in natural habitats that are transformed into artificial habitats, whether this is grassland species that persist in cropland or forest species that persist in degraded forest. However, there are also multiple examples of species generally associated with natural forest habitat that also occur in grassland and multiple artificial habitats (e.g., *Apodemus sylvaticus* and *Peromyscus maniculatus*). Plenty of species are mainly associated with grasslands but also occur in forests and multiple artificial habitats (e.g., *Microtus agrestis* and *M. arvalis*). See also Fig. 3 and Supplementary

Table 1. However, this multi-habitat use mostly applies to generalist species, which we show are also overrepresented among reservoir species (Table 2). We have revised the manuscript to make this aspect clearer (lines 125-129 in the revised manuscript).

14. Lines 130 – Not sure what to think about these descriptive stats – without knowing the methods, at this point, on first reading I am wondering how representative these proportions are.

Our response:

We now give the details of our models (lines 296-305 in the revised manuscript).

15. Lines 158 – This is somewhat vague and the reasoning is not clear to me. Do artificial habitat have more hosts? Is the in the community pool or local level? I am not convinced that is considering community filtering processes. Does high density mean fluctuating populations?

Our response:

We are afraid that we don't fully understand the comment of the reviewer. However, we altered our wording to be more specific and now refer more clearly to the two relevant figures, and we also state that the pattern we found likely applies irrespective of the local species pool (lines 151-155 in the revised manuscript).

16. Lines 177 – “underscore particular pathways to spillover transmission” – this is very vague. Is there definitely a link between population fluctuations and transmission risk? I would suggest sticking to what you are testing and focus talk about reservoir status, which is important in itself.

Our response:

Our study goes beyond determining reservoir status (see our reply to comment #8 above). In a first step, we identified predictors of reservoir status. Here, synanthropy was the most important predictor (Table 1, Fig. 2). In a next step, we quantified determinants of synanthropy as a strong proxy of transmission risk (Table 3-4, Fig. 2). In the latter analysis, population fluctuations are an important predictor of synanthropy (Table 3, Fig. 2; See also our response to comment #8). We think that the transparency with which we have evaluated the strength of our inferences concerning pathways of spillover transmission will allow readers to evaluate for themselves both the strengths of this inference, as well as the data sources underlying the patterns we describe here.

17. Lines 179 – How exactly? What is strength of the relationship? How should we be interpreting this information?

Our response:

We extended the discussion by including “To better mitigate disease outbreaks, surveillance focusing on reservoir rodents exhibiting large population fluctuations appears to be a promising approach¹³. Surveillance should in addition intensify screening for new zoonotic pathogens and/or new reservoirs in rodents exhibiting large population fluctuations and those being hunted.” (lines 209-213 in the revised manuscript).

18. Lines 282. I am not sure what this figure adds extra from the text.

Our response:

We consider this figure to be important and central for guiding both expert and non-expert readers through the different factors that increase transmission risk for rodents in particular. We reference this figure multiple times throughout the manuscript, and we found it a helpful tool to transparently communicate our conceptual framework and our interpretations of these various pathways.

19. Lines 300. What are the grey areas on this figure? 95% CI? If so, the variation in panel b is very broad and needs to be mentioned in the text.

Our response:

We now added in the legend of Fig. 2 that the shaded area shows 95 % CI (lines 524 in the revised manuscript). We also added in the main text that the relationship is associated with high variability (lines 105-107 in the revised manuscript)

20. Lines 306: Not sure what panel B is telling us, can you say more Do proportionally more grassland species go to arable for instance – it all looks fairly proportional to size of the groups. Also, we don't actually know what species move where in reality.

Our response:

Figure 3b provides important information on potential rodent movements from natural habitat and spatial persistence, once natural habitat is converted into artificial one – information that is not given in Fig. 3a. While the large number of rodent species studied does not lend itself to easily presenting each species' movement/persistence, Supplementary Table 1 provides detailed information on the habitats each species occurs in. Here, it is reasonable to assume that species occurring in natural forests and degraded forest can move from the natural forest into degraded forest or that such species may even show site fidelity during habitat transformation. See also lines 125-129 in the revised manuscript.

Reviewer #2 (Remarks to the Author):

Many species of rodents exhibit large scale seasonal or multi-annual population fluctuations. Rodents occupy a variety of habitats, ranging from natural to human-modified habitats, and also serve as reservoir hosts for a variety of pathogens. This manuscript evaluates whether or to what extent the degree of population fluctuations, human impact on the habitat or degree of exploitation of rodents by humans influences spillover of zoonotic diseases. Using complemented existing database of reservoir rodents with 154 non-reservoir rodent species, and also compiled data on s-index which is a commonly used measure of the degree of population fluctuations. They show that zoonotic reservoir status of rodent species is strongly driven by synanthropy, human exploitation and the degree of population fluctuations. Rodents that occupy human-modified habitats, are vulnerable to human exploitation and exhibit large scale population fluctuations are significantly more likely zoonotic reservoirs.

The manuscript is clear, concise and easy to read. Statistical analysis and presentation of results seems adequate. Main conclusions of the manuscript appear to be supported by the data/analyses.

1. I urge the authors to make sure that their database is up to date, and relevant works are acknowledged/cited. For example, there has been a fair amount of work involving tularemia in a cyclic population of common voles in Spanish farmland (e.g., Herrero-Cófreces et al 2021, *Frontiers in Veterinary Science* 8: Article 698454; *Emerging Infectious Diseases* 23 (8), 1377) and plague in rats (Rahelinirina et al. 2021, <https://doi.org/10.1111/1749-4877.12529>). Also, there exists a large body of literature on a community of pathogens in a semi-natural, cyclic vole population in UK (led by M. Begon, X. Lambin, S. Tefler; https://scholar.google.com/citations?hl=en&user=gXVSxRcAAAAJ&view_op=list_works&sortby=pubdate).

Our response:

We very much appreciate the reviewer's positive views on our paper. Upon submission to Nature, with subsequent transfer (without peer review) to Nature Communications, our database was up-to-

date. The papers mentioned by the reviewer above were published after we finalized our search (October 2020). In addition, we screened systematically for quantitative data from the literature with the search criteria described in Methods/Datasets. It is intrinsic to the type of study we did that a dataset will be at least slightly out-of-date immediately after the end date of search due to constantly new studies being published. We would therefore prefer not to continually update our database at this point. To redo our analyses in 5-10 years would certainly be meaningful, but at this point, we would not expect more than quite modest additional contributions to have appeared. The data compilation that we did for this study took us three years, without accounting for the statistical analyses. In addition, the studies from 2021 the reviewer refers to concern *Microtus arvalis*, *Rattus rattus* and *Mus musculus*, i.e., all species that are extensively included in our study and for which we have found much information on reservoir status, habitat preferences, synanthropy, population fluctuations etc. We hope that the reviewer will agree that we have compiled a complete and up-to-date dataset that provides reliable data on the variables that we included in our analyses (cf. Supplementary Table 1).

2. Finally, recent synthesis on rodent population fluctuations and the role of diseases may be relevant (e.g., C. J. Krebs. 2013. Population fluctuations in rodents. Univ. Chicago Press; Oli, M. 2019. Population cycles in voles and lemmings: state of the science and future directions. Mammal Review 49:226–239.)

Our response:

We are aware of the literature mentioned by the reviewer, and we now cite it (lines 26-27 and 173-175 in the revised manuscript). Although both sources describe the potential effects of pathogens and disease on population dynamics of rodents, they do not address the consequences of those fluctuations to zoonotic disease risk. Hence, we cite them in the context of rodent fluctuations but not zoonotic risk.

Reviewer #3 (Remarks to the Author):

1. Overall, this is a great paper and will help to identify potential rodent reservoirs and habitat types that relate to zoonosis. I have a few comments.

Our response

We appreciate these kind words.

2. Line 78-81: It would be useful to state your specific hypotheses

Our response:

We now added specific hypotheses (lines 69-75 in the revised manuscript).

3. Line 110-112: Is there a way to quantify underrepresents or overrepresented. Like how much more than expected?

Our response:

By applying a chi-square test, we quantified the under- and overrepresentation, respectively. We also refer to the patterns evident in Fig. 3. See lines 112-123 in the revised manuscript, where we also added an additional reference to Fig. 3.

4. Line 339-341: could you give an example of a particular search?

Our response:

As we describe in the text, “we combined search strings for topics on countries, species (including their synonyms according to IUCN) and information on population dynamics” (lines 230-232 in the revised manuscript). “The search string for each continent is represented as follows: “TS=(country) and TS=(species) and TS=(abundance or density or population dynamics or amplitude or cycl*)”, with “TS” representing “Topic”, “country” being a list of all countries per continent and “species” being a list of all known rodent species (incl. their synonyms) occurring in the respective continent”. We added this information in the method section (lines 234-238 in the revised manuscript).

5. Line 393-395: I’m not sure what you mean about each predictor values was first tested for its association with the response variable. What type of analysis was this? What is the rationale for using a p value < 0.15 to choose which predictor values were included in the full model. Usually you don’t mix AIC and P values, so is there support for this methods. Also, did you test to ensure there were no correlated predictor variables?

Our response:

Testing by bivariate relationships between candidate predictors and the response variable is a common procedure in epidemiological research and data analysis, especially when there are categories of predictors, which in this work include life-history traits, demographic characteristics, and environmental variables (see examples below). For this pre-selection, we used a generalized linear model (line: 289-292) and relatively non-stringent p-value requirement of p-value <0.15 and we did not use AIC at this stage.

For the full model, we tested for high levels of correlation among predictors using the Variance Inflation Factor (VIF) and a cut-off value of 10. However, VIF did not exceed 6 in our models (line:309-313), and we are thus confident that correlation among predictor variables is not an issue in our models.

Examples of bivariate tests prior to fitting full model:

Nery N, Jr., Aguilar Ticona JP, Gambrah, C, Doss-Gollin S, Aromolaran A, Rastely-Ju´nior V, et al. (2021) Social determinants associated with Zika virus infection in pregnant women. *PLoS Negl Trop Dis* 15(7): e0009612. <https://doi.org/10.1371/journal.pntd.0009612>

Victora CG, Huttly SR, Fuchs SC, Olinto MT (1997) The role of conceptual frameworks in epidemiological analysis: a hierarchical approach. *Int J Epidemiol* 26: 224–227.

Felzemburgh RDM, Ribeiro GS, Costa F, Reis RB, Hagan JE, et al. (2014) Prospective Study of Leptospirosis Transmission in an Urban Slum Community: Role of Poor Environment in Repeated Exposures to the *Leptospira* Agent. *PLoS Negl Trop Dis* 8(5): e2927. doi:10.1371/journal.pntd.0002927

Reviewer comments, second round

Reviewer #3 (Remarks to the Author):

The authors have addressed all of my comments in the revision to my satisfaction.

Reviewer #4 (Remarks to the Author):

I have read with great interest the revised version of this manuscript and I would like to congratulate the authors for the work they have done to reply carefully to comments of previous reviewers. I agree with them about the clarity of the manuscript and the quality of the statistical analyses.

I have just a final minor comment that the authors may take into account. The authors did not discuss the potential role of (global) invasive rodents that may alter the population dynamics of native rodents. The effects could be direct or indirect as showed by Fukasawa et al. (2103) among other studies <https://pubmed.ncbi.nlm.nih.gov/24197409/>
It would be worth to just mention this in the discussion, but I let the authors decide.

Reviewer #5 (Remarks to the Author):

As requested by the editors, I will focus my feedback predominantly on the question of whether the concerns raised by the previous Reviewer 1 have been adequately addressed. I very much agree with that reviewer that the dataset is interesting and carefully collected, and that the questions the authors are attempting to address here are difficult but both worthwhile and relevant to understanding disease risk. However, I share their concerns about the problems of the analysis, and I think there are fundamental conceptual and inferential issues that have not been sufficiently addressed even in this revised version.

The authors have made useful and laudable efforts to address the specific statistical issues raised by reviewer 1 (especially point 7). However, these are mostly relatively technical changes for clarity and testing of model assumptions, and the other mainly text-based changes and rebuttals do not address what I (and reviewer 1) see as the core problems affecting the strength of this manuscript's conclusions.

The two major and interlinked conceptual problems are (1) the lack of a clearly-defined causal model that the authors can both use to guide analysis design and interpretation, and that can aid readers in understanding the results, and (2) insufficient adjustment for the confounding effects of sampling bias that I expect may strongly affect the findings if corrected for properly. Both of these problems I think encompass all of the main concerns raised by Reviewer 1. Neither of these issues are unique to this study, and are consistent challenges for disease macroecology, but an emerging body of work (e.g. Wille et al 2021 PLoS Biology; Gibb et al 2022, which the authors cite; and Albery et al 2022 Nature Eco Evo) is highlighting the importance of addressing them in detail.

I think that the authors have the opportunity to really strengthen their analyses here, and with substantial revisions to the analysis, the paper could be an interesting contribution to knowledge around rodents and zoonotic risk (with the usual caveats of studies at this scale). However, without addressing these fundamental issues, I unfortunately feel that the authors' conclusions are still not convincingly supported by the data and evidence presented here.

(1) The need for a causal model of relationships among drivers

Analysing drivers of zoonotic risk at the species-level is often necessary because of data gaps, but

challenging because the species-level characteristics we are often most interested in (e.g. population size fluctuations, reservoir status, pathogen richness, synanthropy) can be much more geographically variable and linked to local ecologies, than many morphological or life history traits. For example, population fluctuations may be very different in one part of a species range (e.g. in a highly agricultural area with seasonal resource availability linked to human cropping cycles) than another (e.g. a fallow or grassland area with very different seasonal characteristics). This issue is similar for realized pathogen richness, or even synanthropy (e.g. fruit bats becoming increasingly urban to adapt to rapid warming in some areas). The causal relationships between these factors may also differ in different places, or even involve feedbacks: for example, synanthropy might drive population fluctuations (due to e.g. agricultural system seasonality) while simultaneously population fluctuations may drive synanthropy (by causing rodents to enter homes to seek food). Furthermore, our ability to robustly define these characteristics at the species-level depends fundamentally on geographical and taxonomic patterns of sampling effort.

It is impossible to incorporate such spatial complexity in models fitted at species-level. However, all of these complexities will cause confounding relationships among all the variables that fundamentally affect inference, and may make the results very sensitive to how the models are defined. This is one reason why, as Reviewer 1 pointed out, a stepwise or programmatic model selection procedure feels inappropriate for this analysis, which seeks to provide causal (rather than predictive) explanations.

To address this issue and formalise hypotheses, I think what is firstly needed is a clear, visualized causal model (i.e. a formal DAG) of the hypothesized causal relationships of each covariate to the outcome, and importantly, of the relationships among all the covariates. Although the conceptual framework in Figure 1 is a simplified schematic of this, it does not factor in some of the key issues that are central to inference, here: e.g. it does not articulate that reservoir status is being treated as a proxy for transmission risk (which remains unmeasured), or that causality between some variables may be bidirectional, or that different variables may be differentially affected by sampling effort.

The causal diagram can then be used to guide model design to test the key hypotheses of this paper, which the title implies is, "are population fluctuations associated with increasing zoonotic risk?". This will include the choice of which covariates to include based on expected confounding relationships (rather than selecting programmatically).

Such a causal model can also usefully inform broader decisions about how to structure the models and adjust for sampling effort. At the moment, the argument presented by the authors is pretty weak: they argue that synanthropy is associated with zoonotic risk (proxied by host status) and that population fluctuations are associated with synanthropy, and therefore population fluctuations are associated with zoonotic risk (title, lines 160-161 and 198-201; despite statistical evidence in Table 2 suggesting that fluctuations do not clearly differ between host and non host groups). Something like a path analysis (see Albery et al 2022 Nat Eco Evo) may be a far more appropriate tool to actually tackle this question, as it would allow the authors to embed these causal assumptions within the modelling framework itself. It would also allow for a more robust adjustment for sampling bias on multiple covariates (see below).

(2) Adjustment for sampling bias

As Reviewer 1 highlighted several times, the issue of biases in sampling effort could feasibly have an extremely large confounding effect on the results here, since effort will simultaneously affect all 3 core variables in this study (s-index, reservoir status/pathogen diversity and synanthropy). The similarity of Figures 4b, 4d, 4e and 4f to me is strongly suggestive that global patterns of study effort on rodents are biasing all of these variables. Currently, as far as I can tell, sampling effort is adjusted for in these analyses in only a cursory way, via weighting the binomial model using number of studies used to derive the s-index, which I am not convinced makes much sense (although apologies if I have missed more detailed efforts to adjust for effort). This is, to my mind, not robust enough.

At a minimum, I think a species-wide proper proxy for effort is required (e.g. total number of publications and/or pathogen-related publications from PubMed), as an additional covariate to

adjust for effort in the models (and could be usefully mapped in Figure 4 to show how it maps onto the other variables).

However, since sampling effort is likely to affect both the response and predictor variables, simply including publications as an additional fixed effect may not fully deal with the confounding issues. A path analysis might again be a better way to address this issue, by incorporating the effect of publications on both predictor and response variables.

Additionally, I agree with Reviewer 1 that some sensitivity analyses would be very appropriate here to disentangle how effort may be affecting the results – in particular, running the models only including species from the well-sampled regions of the world (e.g. Europe + the Americas). These could be presented in the appendices, but I think – in addition to improving adjustments for effort – they would really strengthen the case for the findings (or highlight their limitations).

Other comments

(1) A slightly smaller but not inconsequential issue is that the s-index, and the concept of population fluctuations as a species-level characteristic, should be more clearly explained and defined. To me, it seems that measured population fluctuations are emergent phenomena arising from interactions between species reproductive traits and local geographical and ecological context, and so tricky to define strictly as an empirical species-level characteristic. Furthermore, looking at the dataset itself, the number and geographical range of studies used to derive the s-index unsurprisingly differs widely between species. Since population fluctuations vary between different parts of a species range, it is difficult to ascertain how much measurement error this might introduce into the s-index between species and areas, and to understand how this might affect the results. For example, better studied European species might have a much more accurate s-index than an African species with only 1 study.

At a minimum, these conceptual and methodological challenges should be discussed in some depth throughout, and a table and summary statistics (or maps) should be provided somewhere showing how many studies were used to calculate this index.

REVIEWER COMMENTS

Reviewer #3 (Remarks to the Author):

The authors have addressed all of my comments in the revision to my satisfaction.

Our response:

We thank the reviewer for the positive response.

Reviewer #4 (Remarks to the Author):

I have read with great interest the revised version of this manuscript and I would like to congratulate the authors for the work they have done to reply carefully to comments of previous reviewers. I agree with them about the clarity of the manuscript and the quality of the statistical analyses.

I have just a final minor comment that the authors may take into account. The authors did not discuss the potential role of (global) invasive rodents that may alter the population dynamics of native rodents. The effects could be direct or indirect as showed by Fukasawa et al. (2103) among other studies <https://pubmed.ncbi.nlm.nih.gov/24197409/>

It would be worth to just mention this in the discussion, but I let the authors decide.

Our response:

We thank the reviewer for the positive response and the remark on the role of invasive rodent species. We now added a section on the complex responses potentially induced by invasive rodent species in the discussion of the revised manuscript (l. 244-247).

Reviewer #5 (Remarks to the Author):

As requested by the editors, I will focus my feedback predominantly on the question of whether the concerns raised by the previous Reviewer 1 have been adequately addressed. I very much agree with that reviewer that the dataset is interesting and carefully collected, and that the questions the authors are attempting to address here are difficult but both worthwhile and relevant to understanding disease risk. However, I share their concerns about the problems of the analysis, and I think there are fundamental conceptual and inferential issues that have not been sufficiently addressed even in this revised version.

The authors have made useful and laudable efforts to address the specific statistical issues raised by reviewer 1 (especially point 7). However, these are mostly relatively technical changes for clarity and testing of model assumptions, and the other mainly text-based changes and rebuttals do not address what I (and reviewer 1) see as the core problems affecting the strength of this manuscript's conclusions.

The two major and interlinked conceptual problems are (1) the lack of a clearly-defined causal model that the authors can both use to guide analysis design and interpretation, and that can aid readers in understanding the results, and (2) insufficient adjustment for the confounding effects of sampling bias that I expect may strongly affect the findings if corrected for properly.

Both of these problems I think encompass all of the main concerns raised by Reviewer 1. Neither of these issues are unique to this study, and are consistent challenges for disease macroecology, but an emerging body of work (e.g. Wille et al 2021 PLoS Biology; Gibb et al 2022, which the authors cite; and Albery et al 2022 Nature Eco Evo) is highlighting the importance of addressing them in detail.

I think that the authors have the opportunity to really strengthen their analyses here, and with substantial revisions to the analysis, the paper could be an interesting contribution to knowledge around rodents and zoonotic risk (with the usual caveats of studies at this scale). However, without addressing these fundamental issues, I unfortunately feel that the authors' conclusions are still not convincingly supported by the data and evidence presented here.

Our response:

These first comments of Reviewer #5 above (modelling and sampling bias) reflect the major comment #1 and #2 below, and we therefore address these comments in our responses below.

(1) The need for a causal model of relationships among drivers

Analysing drivers of zoonotic risk at the species-level is often necessary because of data gaps, but challenging because the species-level characteristics we are often most interested in (e.g. population size fluctuations, reservoir status, pathogen richness, synanthropy) can be much more geographically variable and linked to local ecologies, than many morphological or life history traits. For example, population fluctuations may be very different in one part of a species range (e.g. in a highly agricultural area with seasonal resource availability linked to human cropping cycles) than another (e.g. a fallow or grassland area with very different seasonal characteristics). This issue is similar for realized pathogen richness, or even synanthropy (e.g. fruit bats becoming increasingly urban to adapt to rapid warming in some areas). The causal relationships between these factors may also differ in different places, or even involve feedbacks: for example, synanthropy might drive population fluctuations (due to e.g. agricultural system seasonality) while simultaneously population fluctuations may drive synanthropy (by causing rodents to enter homes to seek food). Furthermore, our ability to robustly define these characteristics at the species-level depends fundamentally on geographical and taxonomic patterns of sampling effort.

It is impossible to incorporate such spatial complexity in models fitted at species-level. However, all of these complexities will cause confounding relationships among all the variables that fundamentally affect inference, and may make the results very sensitive to how the models are defined. This is one reason why, as Reviewer 1 pointed out, a stepwise or programmatic model selection procedure feels inappropriate for this analysis, which seeks to provide causal (rather than predictive) explanations.

To address this issue and formalise hypotheses, I think what is firstly needed is a clear, visualized causal model (i.e. a formal DAG) of the hypothesized causal relationships of each covariate to the outcome, and importantly, of the relationships among all the covariates. Although the conceptual

framework in Figure 1 is a simplified schematic of this, it does not factor in some of the key issues that are central to inference, here: e.g. it does not articulate that reservoir status is being treated as a proxy for transmission risk (which remains unmeasured), or that causality between some variables may be bidirectional, or that different variables may be differentially affected by sampling effort. The causal diagram can then be used to guide model design to test the key hypotheses of this paper, which the title implies is, “are population fluctuations associated with increasing zoonotic risk?”. This will include the choice of which covariates to include based on expected confounding relationships (rather than selecting programmatically).

Such a causal model can also usefully inform broader decisions about how to structure the models and adjust for sampling effort. At the moment, the argument presented by the authors is pretty weak: they argue that synanthropy is associated with zoonotic risk (proxied by host status) and that population fluctuations are associated with synanthropy, and therefore population fluctuations are associated with zoonotic risk (title, lines 160-161 and 198-201; despite statistical evidence in Table 2 suggesting that fluctuations do not clearly differ between host and non host groups). Something like a path analysis (see Albery et al 2022 Nat Eco Evo) may be a far more appropriate tool to actually tackle this question, as it would allow the authors to embed these causal assumptions within the modelling framework itself. It would also allow for a more robust adjustment for sampling bias on multiple covariates (see below).

Our response:

We have now reanalysed our data with structural equation modelling (SEM, also known as path analysis) (l. 354-389 in the revised manuscript) as suggested by the reviewer and included the results in the manuscript (Fig. 2 in the revised manuscript and l. 97-133). We kept the GLMMs (Table 1, 3) to confirm the results of the SEM and to account for the effect of species family on the SEM results and thus correlation among species within the same family, since it is not feasible to include family as a grouping factor in SEM models. For illustrative reasons, we also kept the figure associated with the GLMMs (Fig. 3 in the revised manuscript). The SEM confirmed the importance of synanthropy in affecting reservoir status and that synanthropy, in turn, is influenced by habitat generalism and high population fluctuations, among other factors.

(2) Adjustment for sampling bias

As Reviewer 1 highlighted several times, the issue of biases in sampling effort could feasibly have an extremely large confounding effect on the results here, since effort will simultaneously affect all 3 core variables in this study (s-index, reservoir status/pathogen diversity and synanthropy). The similarity of Figures 4b, 4d, 4e and 4f to me is strongly suggestive that global patterns of study effort on rodents are biasing all of these variables. Currently, as far as I can tell, sampling effort is adjusted for in these analyses in only a cursory way, via weighting the binomial model using number of studies used to derive the s-index, which I am not convinced makes much sense (although apologies if I have missed more detailed efforts to adjust for effort). This is, to my mind, not robust enough.

At a minimum, I think a species-wide proper proxy for effort is required (e.g. total number of publications and/or pathogen-related publications from PubMed), as an additional covariate to adjust for effort in the models (and could be usefully mapped in Figure 4 to show how it maps onto the other variables).

However, since sampling effort is likely to affect both the response and predictor variables, simply including publications as an additional fixed effect may not fully deal with the confounding issues. A path analysis might again be a better way to address this issue, by incorporating the effect of publications on both predictor and response variables.

Additionally, I agree with Reviewer 1 that some sensitivity analyses would be very appropriate here to disentangle how effort may be affecting the results – in particular, running the models only including species from the well-sampled regions of the world (e.g. Europe + the Americas). These could be presented in the appendices, but I think – in addition to improving adjustments for effort – they would really strengthen the case for the findings (or highlight their limitations).

Our response:

From Web of Science, we now carefully extracted for each species the number of available studies on a) population fluctuations, b) zoonotic diseases the species can spread and c) population fluctuations or zoonotic diseases (see Methods l. 314-322). Based on the results from the analyses, we have now added a separate section in the Results on sampling bias (l. 193-209) and discussed potential biases in more detail (now l. 247-250 in the revised manuscript). The number of studies per species are now included in the SEM. The results on the relationship between the number of studies and number of zoonoses harboured by a species are now included in Supplementary Figure 1 and those on geographic bias in Supplementary Figure 2. As we now describe, any effects of sampling bias are likely to be complex and uneven, and our thorough analyses provide for readers a transparency in this regard that we hope will satisfy the reviewer. Despite crude correlations between sampling effort and pathogen detection, we find that a substantial number of species with low sampling effort are well-represented as reservoirs, and others with high sampling effort host few zoonotic pathogens, irrespective of their geographic locations. We thank the reviewer for urging us to set this new standard of transparency for future studies of potential impacts of sampling bias.

Other comments

(1) A slightly smaller but not inconsequential issue is that the s-index, and the concept of population fluctuations as a species-level characteristic, should be more clearly explained and defined. To me, it seems that measured population fluctuations are emergent phenomena arising from interactions between species reproductive traits and local geographical and ecological context, and so tricky to define strictly as an empirical species-level characteristic. Furthermore, looking at the dataset itself, the number and geographical range of studies used to derive the s-index unsurprisingly differs widely between species. Since population fluctuations vary between different parts of a species range, it is difficult to ascertain how much measurement error this might introduce into the s-index between species and areas, and to understand how this might affect the results. For example, better studied European species might have a much more accurate s-index than an African species with only 1 study.

At a minimum, these conceptual and methodological challenges should be discussed in some depth throughout, and a table and summary statistics (or maps) should be provided somewhere showing how many studies were used to calculate this index.

Our response:

We now explained how the s-index is calculated (Methods, l. 308). Indeed, causes of population fluctuations vary among species and might also vary within the distribution range of a species.

However, in our study, we are not studying the causes, but are interested in the consequences of population fluctuations. The accuracy of the s -index can be discussed but does not necessarily increase with the number of studies. Here, it might rather be the length of a time series that determines accuracy. In fact, in our study, there are species from regions that are generally considered as understudied, that have some of the longest time series, with highly reliable estimates of population size and hence s -index. One of these species includes for example the natal multimammate mouse (*Mastomys natalensis*) that exclusively occurs in Africa.

The number of studies included for calculating the s -index is provided in Supplementary Table 1, which was included in the original submission. In this table, we transparently present all data we gathered and also provide information on the length of time series and country of study.

Reviewer comments, third round

Reviewer #5 (Remarks to the Author):

The authors have made fantastic efforts to respond to my previous comments, incorporating a structural equation model approach, clearer description of hypotheses and a fuller adjustment and reporting for sampling bias. I enjoyed reading the new version of the manuscript, I think the results are clearer and more interpretable, and with these analytical adjustments I think the paper should be a nice contribution to the literature on rodent-borne zoonoses at the macro-scale.

Before the paper is published I do however feel that, in light of these revised analyses, it would strongly benefit from the authors revisiting and reframing some of the findings in the text and title. To me these new analyses clearly indicate that it is synanthropy, not population fluctuations, that is most strongly explanatory of rodent reservoir status at the species-level. Indeed, when accounting in the SEM for the mediating influence of synanthropy, the hypothesised causal link between population fluctuations and reservoir status is non-significant and removed from the final model (lines 101-105).

My concern is that the authors argue throughout (e.g. the title; abstract; lines 143-146) that population fluctuations are a driver of reservoir status, including stating that "population fluctuations and associated synanthropy are robust indicators of reservoir status" (line 256). The logic for this argument is that the analysis shows a pathway of fluctuations -> synanthropy -> reservoir status. However, the path analysis coefficients suggest this causal chain is rather weakly coupled overall, such that if we were only provided information about a poorly-known species' s-index, this would not be particularly predictive of its reservoir status, in the absence of additional information about whether it is synanthropic.

As a result I feel that the manuscript currently overstates the significance of population fluctuations to reservoir status, via the use of this logic throughout the results and discussion, and especially in the title of the paper – which states strongly that population fluctuations drive 'transmission risk' (despite reservoir status itself only being an imperfect proxy for the latter). So I think the authors should strongly consider changing these to better reflect the evidence presented. To me, the results and methodology are interesting enough (e.g. in showing the network of associations between body mass, population dynamics, landscape and human interactions, and reservoir status) to stand by themselves without over-reaching.

REVIEWERS' COMMENTS

Reviewer #5 (Remarks to the Author):

The authors have made fantastic efforts to respond to my previous comments, incorporating a structural equation model approach, clearer description of hypotheses and a fuller adjustment and reporting for sampling bias. I enjoyed reading the new version of the manuscript, I think the results are clearer and more interpretable, and with these analytical adjustments I think the paper should be a nice contribution to the literature on rodent-borne zoonoses at the macro-scale.

Our response:

We thank the reviewer for the positive response.

Before the paper is published I do however feel that, in light of these revised analyses, it would strongly benefit from the authors revisiting and reframing some of the findings in the text and title. To me these new analyses clearly indicate that it is synanthropy, not population fluctuations, that is most strongly explanatory of rodent reservoir status at the species-level. Indeed, when accounting in the SEM for the mediating influence of synanthropy, the hypothesised causal link between population fluctuations and reservoir status is non-significant and removed from the final model (lines 101-105).

My concern is that the authors argue throughout (e.g. the title; abstract; lines 143-146) that population fluctuations are a driver of reservoir status, including stating that “population fluctuations and associated synanthropy are robust indicators of reservoir status” (line 256). The logic for this argument is that the analysis shows a pathway of fluctuations -> synanthropy -> reservoir status. However, the path analysis coefficients suggest this causal chain is rather weakly coupled overall, such that if we were only provided information about a poorly-known species' s-index, this would not be particularly predictive of its reservoir status, in the absence of additional information about whether it is synanthropic.

As a result I feel that the manuscript currently overstates the significance of population fluctuations to reservoir status, via the use of this logic throughout the results and discussion, and especially in the title of the paper – which states strongly that population fluctuations drive ‘transmission risk’ (despite reservoir status itself only being an imperfect proxy for the latter). So I think the authors should strongly consider changing these to better reflect the evidence presented. To me, the results and methodology are interesting enough (e.g. in showing the network of associations between body mass, population dynamics, landscape and human interactions, and reservoir status) to stand by themselves without over-reaching.

Our response:

The concerns raised by the reviewer above relate to the importance of population fluctuations (s-index) versus synanthropy for explaining reservoir status. We agree with the reviewer that we

should avoid identifying population dynamics as the main driver. Indeed, synanthropy is the most important explanatory variable. Our study identifies synanthropy as a defining characteristic of nearly all (95%) currently known rodent reservoirs. However, synanthropy of rodents is not a dichotomic variable (yes/no). In fact, only six rodent species are known as truly synanthropic species (*Bandicota bangalensis*, *Mastomys natalensis*, *Mus musculus*, *Rattus exulans*, *R. norvegicus*, and *R. rattus*; Supplementary Data 1). All other of the 155 synanthropic rodent species are only occasionally synanthropic (Supplementary Data 1), which implies that they move into human dwellings only sporadically. Population dynamics of rodents are important for explaining this synanthropic behaviour. The propensity of rodent species to show occasional synanthropy increases with their tendency to display pronounced population fluctuations (see Table 3, Fig. 3b). Hence, population fluctuations are an important determinant of synanthropy, with occasionally synanthropic species demonstrating this behaviour likely, but for still largely understudied reasons, during periods of high population abundance/density. Our study therefore supports the role of both population fluctuations and synanthropy as determinants of reservoir status and transmission risk. We acknowledge this importance in the revised title (“Population fluctuations and synanthropy explain transmission risk in rodent-borne zoonoses”). In the method section of the abstract, we now explicitly mention that we not only analysed the rodents’ population fluctuations but also their habitat use (l. 21-23). In the result section of the abstract, we consider that there is balance between population fluctuations and synanthropy (l. 24-27). In the results of the main text, we are now explicitly addressing true and occasional synanthropy and added “Of the 155 synanthropic species, only six are considered as truly synanthropic, i.e., dominantly but not exclusively occurring in or near human dwellings, while the remaining species only occasionally show synanthropic behaviour (Supplementary Data 1).” (l. 116-119).